

# Moho topography beneath the Eastern European Alps by global phase seismic interferometry

Irene Bianchi[1,2], Elmer Ruigrok[3,4], Anne Obermann[5], and Edi Kissling[5]

[1]Istituto Nazionale di Geofisica e Vulcanologia, Via di Vigna Murata 605, 00143, Rome, Italy
[2]Institut für Meteorologie und Geophysik, Universität Wien, 1090 Wien Althanstraße 14 (UZA II)
[3]Royal Netherlands Meteorological Institute, De Bilt, The Netherlands
[4]Utrecht University, Utrecht, The Netherlands
[5]Swiss Seismological Service, ETH Zurich, Zurich, Switzerland

**Correspondence:** Irene Bianchi (irene.bianchi@univie.ac.at)

**Abstract.** In this work we present the application of the Global-Phase Seismic Interferometry (GloPSI) technique to a data-set recorded across the Eastern Alps with the EASI temporary seismic network (Eastern Alpine Seismic Investigation). GloPSI aims at rendering an image of the lithosphere from the waves that travel across the core before reaching the seismic stations (i.e. PKP, PKiKP, PKIKP). The technique is based on the principle that a stack of autocorrelations of transmission responses

mimics the reflection response of a medium, and is used here to retrieve information about the crust-mantle boundary, such as its depth and topography. We produce images of the upper lithosphere using 64 teleseismic events. We notice that with GloPSI, we can well image the topography of the Moho in regions where it shows a nearly planar behaviour (i.e. in the northern part of the profile, from the Bohemian massif to beneath the Northern Calcareous Alps). Below the higher crests of the Alpine chain, and the Tauern Window in particular, we cannot find evidence for a typical boundary between crust and mantle. The

GloPSI results indicate the absence of an Adriatic crust made of laterally continuous layers smoothly descending southwards. On the contrary, our results confirm the observations of previous studies suggesting a structurally complex Moho topography and faulted internal Alpine crustal structure.

## 1 Introduction

As part of the Alpine-Himalayan orogen, the European Alps are the result of the subduction of the Alpine Tethys and European

paleomargin beneath the Adriatic microplate and the subsequent continent-continent collision that led to a 200km wide convergence zone with a significant crustal root (e.g. Handy et al., 2015, and references therein). Due to the distinctively different chemistry between the crust and mantle leading to very different physical properties of the crust with respect to the uppermost mantle, the base of the crust denotes a first-order velocity interface and is the seismically best visible subsurface structure in the Earth. First noted by A. Mohorovicic 1910, this velocity interface –called Moho in his honor-,in seismic records from

intra-crustal earthquakes produces a specific set of waves that today are well known in seismology as the direct wave Pg, the wide-angle reflection from the Moho (PmP) and the critically refracted wave Pn travelling in the uppermost mantle along the Moho (e.g. Giese et al., 1976). These three specific seismic phases are well visible also in record sections from blasts and thus



the Moho quickly became the main target of long-range controlled source seismic (CSS) investigations (Prodehl and Mooney, 2012). If well-imaged in refraction seismic record sections, these three phases define the typical triplication that allows to reli-

ably derive the depth of the Moho and the average velocities of the crust and of the uppermost mantle. The CSS methods have successfully been applied in thousands of experiments around the world (Prodehl and Mooney, 2012) to establish thickness and main layering of the crust mostly in continental but also in oceanic lithosphere. With nearly 200 CSS profiles in the greater Alpine region (e.g., Roure et al., 1990; Blundell et al., 1992; Scarascia and Cassinis, 1997; Fantoni et al., 2003; Kissling et al., 2006; Brückl et al., 2007; Hrubcova and Geissler, 2009; Grad et al., 2009), arguably the Alps denote the best studied orogen by

both refraction and near-vertical reflection seismics – the sister CSS method to refraction seismology. In near-vertical reflection seismic profiles along several transects across the Alps (Roure et al., 1990; Pfiffner et al., 1997; TRANSALP Working Group et al., 2002), the Moho has been commonly imaged as a relatively narrow band of high reflectivity (e.g. Holliger and Kissling, 1991), and along the TRANSALP transect this high reflectivity Moho band well correlates with the results obtained by receiver functions for the Moho topography (Kummerow et al., 2004). Anyways, unravelling the Moho beneath the Alps turned out to

be a challenging task (e.g. "The problem of the Moho in the Alps" Laubscher, 1990). After the closure of major and minor oceans, the Alpine Tethys with its several arms and embayments such as the Penninic and the Meliata oceans (e.g. Neubauer et al., 2000), the continental Europe and continental parts of the much smaller plate Adria collided (e.g Handy et al., 2010). For the Eastern Alps, tectonic reconstructions have shown that the convergence between the two plates involved hundreds of kilometres, though there is no consensus on the precise amount of shortening (Rosenberg et al., 2018, and references therein).

Likewise, while there is a general agreement that the European and the Adriatic Moho are offset across the plate boundary in the Alps, the exact Moho topography beneath the Eastern Alps is still a matter of debate. The wide-angle reflection (PmP) from the Moho is a unique and very strong seismic phase in CSS record sections that has successfully been used to map the Moho surface in the Eastern Alpine region (Behm et al., 2007) and to define a Moho gap beneath the core of the Eastern Alps (Bleibinhaus and Brückl, 2006; Spada et al., 2013). In contrast to the "normal" Central European continental crust (Mueller,

1977; Ye et al., 1995), the older Bohemian crust exhibits an anomalously high-velocity lowermost crustal layer well-known from cratonic regions (e.g. Behr et al., 1994; Luosto, 1997; Kozlovskaya et al., 2004). Along the CEL09 profile crossing the Bohemia massif, Hrubcová et al. (2005) documented average upper to lower crustal velocities of 6.0 km/s to 6.4 km/s, similar to the velocity structure for the northern Alpine foreland further W (Ye et al., 1995), yet underlain by an additional crustal layer of anomalously high-velocity of 7.0 km/s regionally varying in thickness up to 12 km and a sub-Moho velocity of 8.0

km/s. Several long-range seismic experiments have been carried out in the Eastern Alpine area. The Alpine longitudinal profile (named ALP75) extended along the axis of the Western and Eastern Alps, reaching the Pannonian basin (Yan and Mechie, 1989; Scarascia and Cassinis, 1997). Moreover, during the years 2000 and 2002, long range CSS experiments, named CEL-EBRATION 2000 and ALP 2002, have covered the area from the Eastern European platform in the north-east to the Adriatic foreland in the south-west (Guterch et al., 2004; Brückl et al., 2003). The temporary dense deployment of passive seismic

stations within the EASI project (Eastern Alpine Seismic Investigation, AlpArray Working Group, 2014; Hetényi et al., 2018) was conceived to add information on the crustal structure and Moho depth, with respect to previous investigations through a set of high-quality seismic data. The temporary EASI array consisted of 55 broadband seismic stations deployed along a



550 km north-south transect from the Bohemian Massif to the Adriatic coast at a Longitude of about 13.4°E (Figure 1). EASI followed the same trajectory as one of the ALP 2002 profiles, namely the Alp01 (Brückl et al., 2007), which extended from the Bohemian Massif to the Adriatic foreland (Figure 1). The trajectory of EASI crosses the lines of two of the previously mentioned active seismic profiles, namely the Cel09 (Hrubcova and Geissler, 2009) and ALP75 (Yan and Mechie, 1989). Both show the Moho depth with low uncertainties. EASI, at 110 km from its Northern edge, crosses the Cel09, according to which the European Moho interface is at 32 km depth; at 375 km from its northern edge, EASI crosses ALP75, which marks the European Moho at 48 km depth.

Most of the information we have about the Moho in the study area are derived from CSS experiments (e.g. Yan and Mechie, 1989; Scarascia and Cassinis, 1997; Waldhauser et al., 2002; Bleibinhaus and Gebrande, 2006; Behm et al., 2007; Brückl et al., 2007; Diehl et al., 2009; Hrubcova and Geissler, 2009; Spada et al., 2013) and few are from passive seismic experiments (e.g Kummerow et al., 2004; Bianchi and Bokelmann, 2014; Hetényi et al., 2018). With the triplication of the typical seismic phases from Moho well visible, CSS refraction seismic methods allow to uniquely identify and model the crust-mantle boundary and to differentiate the Moho from other velocity interfaces, the latter task often being a challenge with passive seismic methods. In CSS experiments though, sources and receivers are both at the surface. With reliable information about the velocities, the depth of a reflecting structure like the Moho can be estimated but not its precise location nor its true dip. The process to restore the true subsurface geometry of reflecting structural elements obtained by seismic profiling is called migration. In a true 3D environment like the Alpine orogen and its forelands though, the migration yields ambiguous results unless many crossing CSS profiles sample the same 3D volume and their 2D results are combined (Kissling et al., 1997) by 3D migration to obtain a Moho map (e.g. Waldhauser et al., 1998). For obvious economic reasons the number of CSS profiles and experiments for academic purposes is limited. This is particularly true in the Eastern Alps, where the CSS profiles mentioned above provide reliable, but very sparse information about the Moho topography that needs to be interpolated and is interpreted with a Moho triple junction (Brückl et al., 2007) or with a Moho gap (Bleibinhaus and Gebrande, 2006). The latter interpretation is strongly supported by Spada et al. (2013) based on critically assessing all available seismic information about the Moho beneath the Alps with regards to their reliability, 3D migration and depth uncertainties and considering the results of PmP mapping presented by Bleibinhaus and Brückl (2006) and Behm et al. (2007). Several attempts were made to image the Moho discontinuity in the Eastern Alps with passive seismic methods exploiting distant earthquakes. Single stations receiver function analysis by both Ps and Sp (Bianchi et al., 2014, 2015) gave scattered values when locating the Moho beneath the higher Alpine crests, suggesting the presence of several seismic discontinuities and anisotropy (Bianchi and Bokelmann, 2014). The interpretation of the RF data set along EASI (Hetényi et al., 2018) shows a clear difference between the signal in the northern part of the profile, where the European Moho is clearly imaged by different approaches, and the southern part of the profile, where the RF results show several features of limited extent and at depth intervals that may correspond to either the lower crust and/or the mantle lithosphere. Near the southern end of the EASI profile, the RF results image the Adria Moho dipping slightly toward N. In conclusion, in the wide central section of the Eastern Alps the Moho is not imaged due to poorly reflective signals (PmP phases from CSS, e.g. Bleibinhaus and Gebrande, 2006; Behm et al., 2007) or poor and inconclusive converted signals by RF (Hetényi et al., 2018). Here, we use seismic interferometry applied to the records

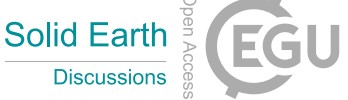

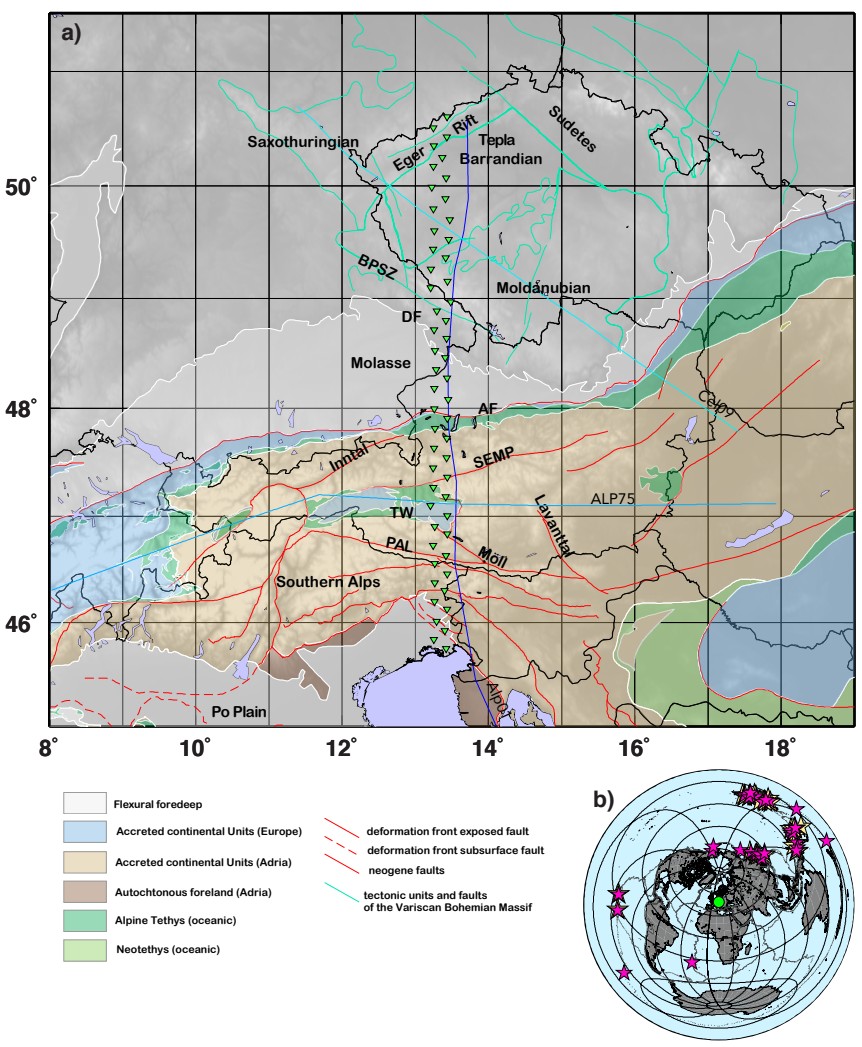

**Figure 1.** a) Map of the wider study area showing the location of the seismic stations (green triangles) and the traces of previous active seismic profiles (ALP75, Cel09, Alp01). Colours on the background correspond to the generalized tectonic map of the Alps (Bigi et al., 1990; Bousquet et al., 2012; Froitzheim et al., 1996; Handy et al., 2010; Schmid et al., 2004, 2008). b) Globe with the location of EASI transect (green) and epicenters of teleseisms used for GloPSI imaging (red stars for the 27 events used for the image in Figure 2, and in yellow the further 37 events used for the image in Figures 4a and S8).





of distant earthquakes, for adding information on the long-debated nature of the lower crust and Moho in this part of the Eastern Alps. The term seismic interferometry refers to the principle of generating new seismic responses of virtual sources (Scuster, 2001) by correlating seismic observations at different receiver locations. Passive seismic measurements are turned into deterministic seismic responses. With global-phase seismic interferometry (GloPSI) we use this principle. We estimate lithospheric-scale reflection responses by autocorrelating and stacking primarily global phases; waves that travel across the core before reaching the seismic stations. Through autocorrelation, a response is obtained that would be measured if there was a co-located source and receiver at the same station. This novel technique has been developed and presented in Ruigrok and Wapenaar (2012), and in the last years was applied for several case studies for imaging the Earth's lithosphere (Nishitsuji et al., 2016; Frank et al., 2014; Van Ijsseldijk et al., 2019). In other implementations with distant seismicity, primarily P-phase correlations are used (Ruigrok et al., 2009; Sun and Kennett, 2016; Pham and Tkalcic, 2017; Tauzin et al., 2019). Based on results from previous applications, we expect that this technique helps identifying the Moho as the boundary between a reflective crust and a less reflective mantle.

## 2 DATA and METHOD

### 2.1 DATA

We collected broadband data from 55 seismic stations belonging to the EASI transect (fully operating between 08/2014 and 08/2015, see Hetényi et al., 2018), now publicly available through EIDA website (https://www.orfeus-eu.org/). We selected earthquakes within the recording time of the EASI deployment at epicentral distances ($\Delta$) between 120° and 180° with M> 5.6. To increase the ray-parameter range, we added events from the northern and southern backazimuthal directions between 70° and 90° $\Delta$, to give an in-line illumination of the profile. After visual inspection, we retain a total of 64 events with high SNR around the P onset (listed in Table T1). We have used PKIKP phases (events from $\Delta$ larger than 120°) which arrive subvertically below the seismic stations and are added to this a handful of P- phases from epicentral distances 70°-90°, to improve the imaging of dipping structure. We discarded events occurring around 150° $\Delta$, for which we observe triplications of the P wave (Adams and Randall, 1963) (i.e. we discarded the time windows with multiple dominant phase response). The selected 64 events display a high station coverage. Our study makes use of the records starting at 10s before and ending 80s after the onset of the P-wave.

### 2.2 METHOD

For the computation of the GloPSI images, we follow the steps indicated in Ruigrok and Wapenaar (2012). The P-direct waves reaching the single seismic station are followed by reverberations that reflect at seismic interfaces at depth and reach the receiver again. Following Claerbout (1968) and Wapenaar (2003) the reflection response at the seismic station is achieved by autocorrelating the transmission response, selecting minus the causal result and muting the delta pulse. We repeat the autocorrelation step for varying illumination angles and then stack together the results in order to cancel the spurious phases





created and to enhance the real signals (Snieder, 2004; Ruigrok et al., 2010). For the phases used, the ray parameter varies from
0 to 0.06 s/km, which suffices to retrieve the zero-offset response (i.e. coinciding source and receiver at the surface) in case of
horizontal and gently dipping interfaces. After applying instrument-response deconvolution and bandpass filtering (0.04 to 0.8
Hz), we apply spectral balancing (Bensen et al., 2007), which broadens the band of the signal mitigating the depletion of energy
at the high end of the spectrum due to dissipation, thus balancing the contribution of all spectral frequencies and equalizing it
for the different earthquakes to enhance the stacking later on. Then, we autocorrelate the phase response on the Z component
of each earthquake at each station and repeat the autocorrelation for all events. The autocorrelation of individual events at each
station is stacked to suppress incoherent features and enhance coherent features (e.g. Pham and Tkalcic, 2017, and references
ther). The next steps of the processing include the removal of the delta pulse, a coherent and high amplitude pulse at t=0,
by muting the first few seconds and removing the low-frequency sidelobes with a high-pass filter. Then the application of a
static correction and surface-related multiple elimination (Verschuur and Berkhout, 1997) is done. We test the method using
different sub-ensembles of our selected 64 events, as shown in Figure 2 and in the supplementary material (Figures S1 to S8).
For each sub-ensemble, we produce four panels showing the a) basic amplitude retrieval (BAR), which corresponds to the
stack of autocorrelated traces after spectral balancing; b) the delta pulse removal; c) multiple suppression; d) the same as c) for
actual station distance. The final image of the crust is then depth-migrated using a velocity model obtained from deep seismic
refraction/wide-angle reflection profiling along the Alp01 profile (Bleibinhaus et al., 2004). This refraction profile runs parallel
to the EASI profile providing an estimate of the P- waves velocities of the crust and uppermost mantle for the region between
profile distances 140km and 300km (we show in Figure S10 the P- velocity model and how it compares to other models).

## 3   RESULTS

To avoid geometrical distortions when imaging with a strong reflection-transmission signal, the interface should be planar and
continuous over at least 20 km, which corresponds to the first Fresnel volume in teleseismic waves. Shorter, irregularly dipping
and separated interface sections may result in seemingly consistent reflection-transmission patterns though their geometries are
strongly distorted and some features are of entirely artificial origin (similarly to the smile effects in active reflection seismics,
e.g. Clauser, 2018). Within the lithosphere only the Moho fulfils these requirements, and therefore we will consider the signal
generated only by the Moho for our interpretation. We show in the supplementary text and figures S1 to S8 how the results of
the application of the GloPSI are sensitive to the choice of the pool of events used for imaging (both concerning the spatial
distribution, the magnitude and a balanced number of events on the two sides of the profile) and that including a larger number
of events does not always imply a better resolved image if the selected events have low magnitude or if they are co-located.
When a small number of sources is used, strong horizontal artefacts can be seen over the interferometric result. The origin of
these artefacts are cross-terms between first arrivals and depth phases. In Ruigrok and Wapenaar (2012), only 17 global phases
could be used and a few cross-terms remained visible after applying seismic interferometry. The cross-terms were suppressed
by removing the average over the array, at the cost of also removing real features that are horizontal over a large part of the
array. For EASI, many more global phases are available (64 instead of 17) and no average removal is applied. For the final





result (shown in Figure 2), a subset of 27 phases was selected according to the epicentral location and magnitude. For this selection, we avoided to include clusters, referring to multiple events with epicentres located within 3 degrees in both distance and backazimuth. Out of each cluster, we included the event with the highest magnitude (listed in Table T1 and shown as red stars in Figure 1 inset).

The first order information about the crust-mantle boundary is already present on the basic amplitude retrieval (BAR) image as a strong positive (red) signal (Figures 2a, S7a, S8a). This preliminary result is used as a guide for further interpreting the images obtained from the sequence of processing steps, to avoid interpreting potential artefacts introduced by the processing that might mislead our final interpretation. After the delta pulse removal, which cuts large amplitudes at t=0, the Moho signals are clearly visible (as blue-red-blue triplet, marked in Figure 2b). In this step we also notice the change from a single positive signal (in the BAR panel Figure 2a) to the blue-red-blue phase alternation (DPR image, Figure 2b). Within the GloPSI images, we look for the blue-red-blue triplet (e.g. Ruigrok and Wapenaar, 2012) as marker of a positive impedance contrast (increasing velocity with depth). The Moho is imaged as a triplet signature with a red (positive) signal in the centre and the typical two side lobes of the wavelet creating such characteristic blue-red-blue feature. The produced image shows a very clear separation between crustal reflectivity and mantle reflectivity (clearly seen in the BAR gather, Figure 2a black dotted line). In Figure 2a the BAR amplitudes highlight very well the difference between crustal features (positive, red) and mantle features (negative, blue); this is particularly true for the northernmost stations (1 to 30), while for the southern stations this clear division is lacking (i.e. an alternation of monochromatic signal is observed after 7 seconds in time, Figure 2a, brown dash-dot rectangle). This separation between crustal and mantle reflectivity corresponds to the blue-red-blue triplet visible at the same arrival times, identified as the signature of the positive impedance contrast at the Moho, and which is visible in the other gathers (i.e. DPR and multiple suppression, Figure 2b and 2c respectively, highlighted by dashed lines). The same features are observed and marked in Figures S7 and S8 but in Figure 2 their strength is enhanced thanks to the input data selection. For stations from 37 to 53 a red phase is identified at 12 seconds in the BAR (marked by a brown line in Figure 2a), with a constant arrival time throughout the transect. This phase is likely an artefact arising from an unbalanced number of events from southern backazimuthal directions that we will not interpret. We also computed the GloPSI response for 100 bootstrapped sets of events (for both cases of including 64 and 27 events). The results have been used to estimate the mean and standard deviation of the amplitudes associated with the images, and are shown in Figure 3 for the 27 events and in Figure S9 for the 64 events; the first has higher STD focusing at confined delay times; the second has lower STD but spread for larger delay times. The northernmost 30 stations show smaller standard deviation values, meaning that the autocorrelated traces are more similar to each other for this part of the transect. The time location of larger standard deviation values highlights the phases displaying large variability, that should not be interpreted geometrically.

Figure 4 shows the depth migrated GloPSI results along the EASI profile for (a) the entire dataset (64 teleseismic sources) and (b) for the subset of 27 teleseismic sources. Note that the GloPSI results include clearly visible signals that originate from two different kind of sources: (1) the primary sources of visible wave energy are the selected earthquakes at teleseismic distances (in this study the entire data set includes 64 events and the subset 27 events that are closer clustered in space) and (2) the second source denotes the target structure immediately beneath the station array. The signals of the former sources are



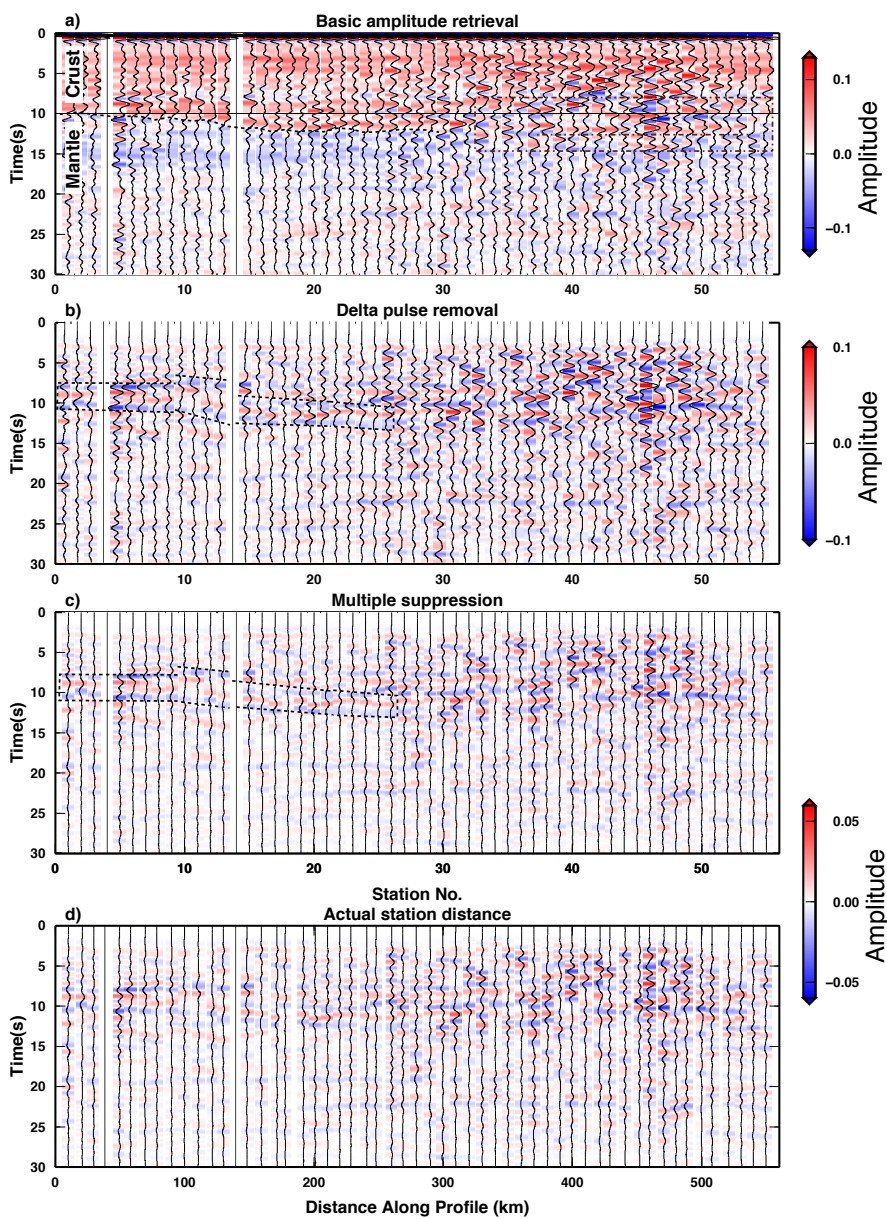

**Figure 2.** Steps of the GloPSI processing on an ensemble of 27 events selected from the pool of events in Table T1. (a) Basic amplitude retrieval, the boundary between crustal and mantle features is marked by a black dashed line, monochromatic signal in the southern part of the profile is highlighted by a brown box, and a constant phase at 12s is marked by a brown dash-dotted line. (b) delta pulse removal, (c) multiple correction and static correction, (d) amplitudes displayed according to the station distance along the N-S direction. Blue-red-blue triplet is outlined between dashed lines in panels b and c.





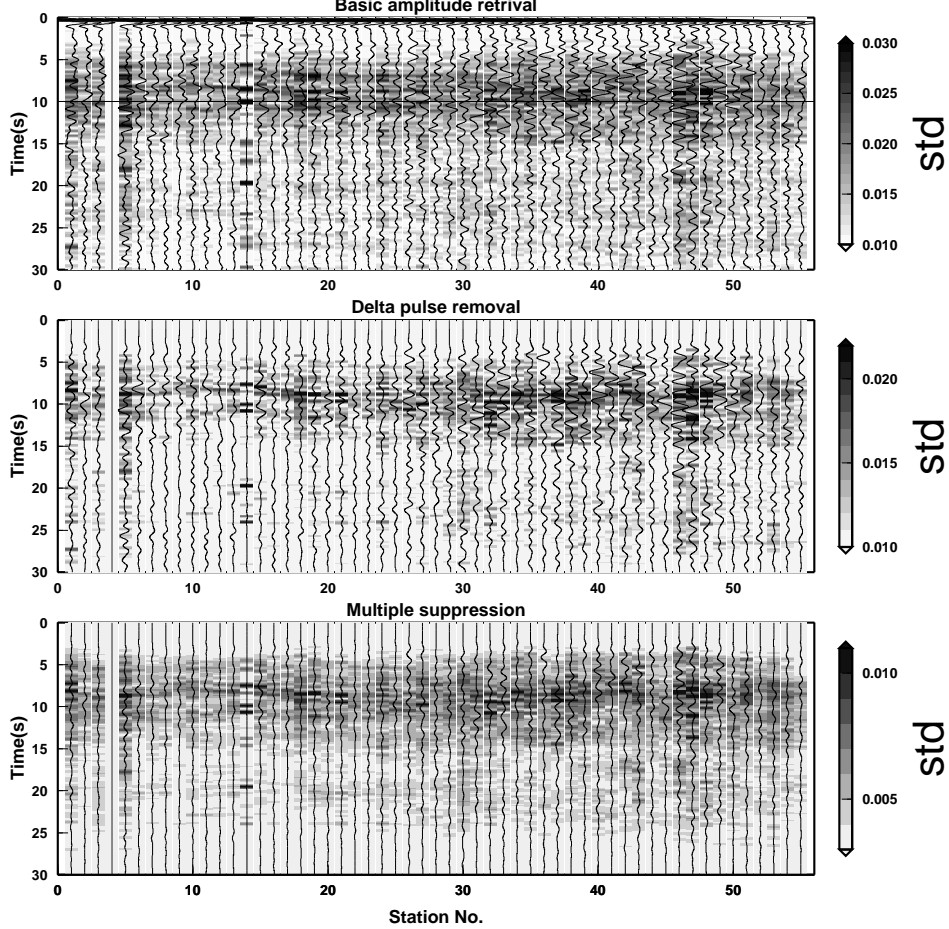

**Figure 3.** Standard deviation calculated over 100 samples generated by bootstrapping events ensembles by the pool of 27 selected events (marked by the asterisk in Table S1). The wiggle on top is the mean wiggles over the 100 bootstrapped wiggles.





called source-side reverberations (SSR) and from the latter receiver-side reverberations (RSR). The comparison between the two results (Figure 4a and b) shows that by including a larger number of events with hypocentres distributed over a larger teleseismic source region, some SSR signals might be reduced in amplitude or locally even cancel out. This is documented by

two marked features running along the entire profile visible at depth between 65 and 85 km (area 2, Figure 4b) and which are disappearing in the northern part of the profile (area 1, Figures 4a and 4b). This shows that adding more phases with different SSRs, simply helps in unveiling the receiver-side structure. This does not happen, though, below the Alpine region (southern part of the profile, area 2 in Figure 4), where the Moho features –as one of the prime targets of the RSR- lose lateral coherency and complex reverberations are imaged. Here, the crust itself exhibits a high structural complexity that prohibits consistent

cross-correlation and migration of deeper structure, such as the Moho. Hence, it is quite likely that the entire or at least most of the signal below the Alps is dominated by artefacts. Consequently, we decide to use the image obtained with 27 events (Figure 4b) and to focus our interpretation on the Moho topography in the northern part of the profile, where we obtained consistent signals.

      In the migrated image in Figure 4b, we pick the lower zero crossing (within the blue-red-blue triplet) in the 0-270 km

section of the profile for the 27 events (Figure 4b). We then smooth this interface for a 50 km Fresnel zone for deriving the Moho topography (Figure 4c). In Figure 4c also marked is a poorly resolved section of the Moho at the southernmost end. While the few lower amplitude sub-horizontal features in the mantle (Figures 4a and 4b, marked by 1 and 2) have been shown to represent SSR (see above), the relatively strong amplitude signals in the crust are nearly identical for either 64 events (Figure 4a) or 27 events (Figure 4b) and, therefore, can only be attributed to reflectivity beneath the receiver array. Note that while

these moderate to strong amplitude signals of relatively short length (up to 50km) above the Moho signal are rather common in GloPSI results (e.g. Ruigrok and Wapenaar, 2012) and while they are visible all along the EASI profile, beneath the Alps (from profile distance 300km to 520km) they dominate the image from the top of the crust to where we would expect the Moho based on previously published CSS data (Yan and Mechie, 1989). The difference in the image of these "crustal features" between profile distance 0km and 300km (Bohemian massif and northern Alpine foreland) and below the Eastern Alpine

orogen, suggest the signals representing at least in parts internal crustal structure. Unfortunately, the 3D crustal structure of the Eastern Alps below 15 km depth is still poorly known (Behm et al., 2007), compared with the well-known crustal structures beneath the Central and Western Alps (e.g. Kissling et al., 2006) and with reference to the tectonic style and geologic evolution of the orogeny (e.g. Willingshofer et al., 2013; Rosenberg and Kissling, 2013, and references therein). However, we can expect a rather complex crustal structure beneath the Eastern Alps, in particular, regarding the lower crust and the crustal root (Handy

et al., 2015). Furthermore, while extrapolations of the Moho topography along the EASI transect across the plate boundary beneath the Alps from CSS profiling (e.g. Brückl et al., 2007) and from RF (Hetényi et al., 2018) differ substantially, they agree in the great complexity of the crust-mantle transition zone. The suggested structural complexity of Moho topography and internal Alpine crustal structure correlates very well with our GloPSI results. If the European Moho continued to descend smoothly toward S beyond profile distance 270km and if it was overlain by a simple crustal structure of laterally continuous

layers, the crust would just increase in thickness and exhibit smooth lateral velocity variations. GloPSI would certainly show this distinct change across the northern Alpine front.



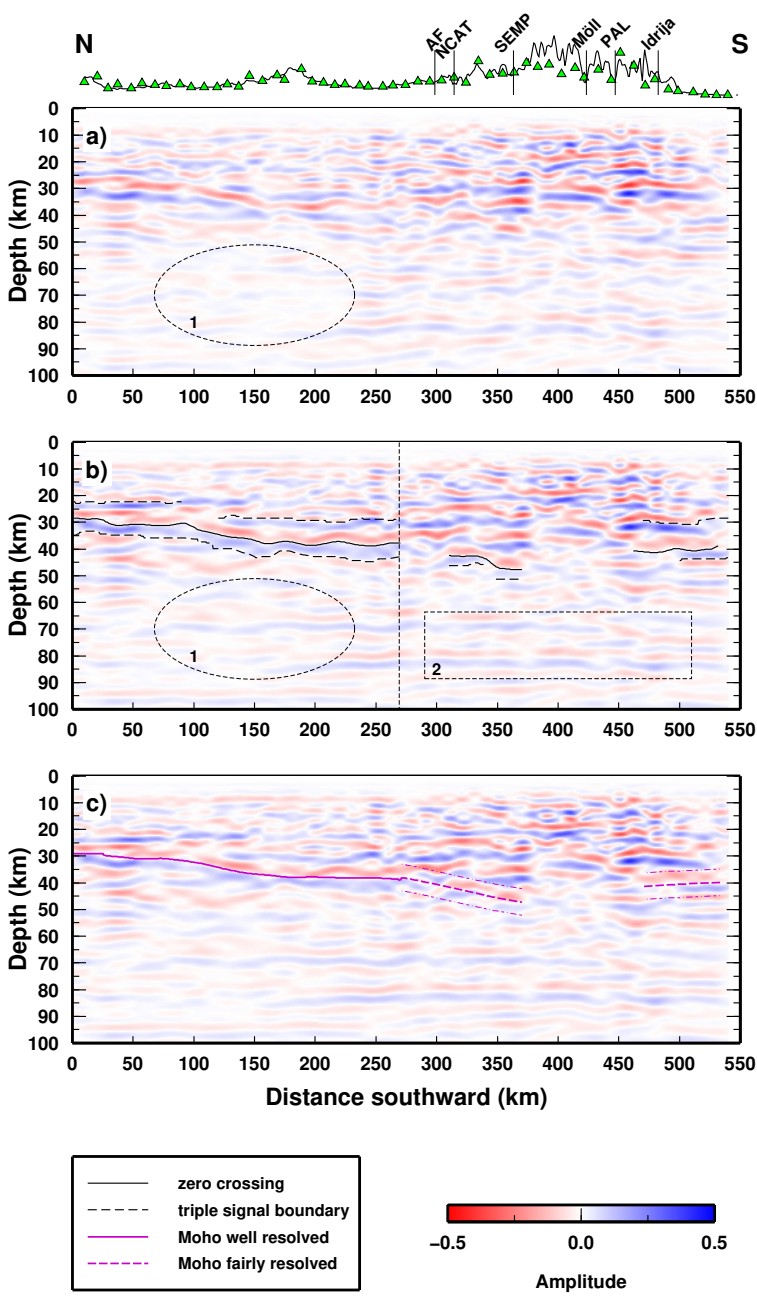

**Figure 4.** Reflectivity images of the crust and upper mantle along EASI; in the background the interpolated figure, in which blue-red-blue triplet marks the presence of a positive interface (i.e. an increasing impedance contrast with depth). a) Depth migrated GloPSI image generated by using 64 events. b) Depth migrated GloPSI image generated by using 27 events, black solid line marks the lower zero crossing within the blue-red-blue triplet, black dashed line marks the upper and lower boundary of the triplet; features 1 and 2 are described in the text. c) background same as (b), solid and dashed purple line show the picked Moho depths reported in Table S2.





## 4 DISCUSSION

The Moho GloPSI results obtained in this study are documented as a migrated image in Figure 4c and compared with published information about the Moho along the EASI transect in Figure 5. In the GloPSI image we notice a clear divide between two

domains along the EASI transect. The northern part of the profile (0 to 270 km, possibly 300km distance along profile, Figure 4b), is characterized by low amplitude reflectors within the crust and one pronounced feature (both in amplitude and length) that can univocally be related to the Moho interface above the uppermost mantle lithosphere that is nearly transparent. The southern part of the profile instead (south of about 300 km), is characterized by high amplitude reflectivity within the whole crust. The observed alternation of positive and negative phases testifies to the presence of a complex velocity structure with

several interfaces of strong velocity discontinuities. In Figure 5 we included the information from several CSS studies (5b and 5d) and from RF (5c) studies. In particular, the CSS profiles analysed by Hrubcova and Geissler (2009) and by Yan and Mechie (1989), are crossing the EASI profile at 110 and 375 km respectively, and they provide two reference points for the Moho depth (stars in Figure 5b, 5d). We compare our image also with the refraction seismic model by Brückl et al. (2007) and the Moho depths from the study of Spada et al. (2013), which combined the published CSS profile results with well-resolved

Moho depths based on PmP wide-angle reflections (from Behm, 2006). The strength of our new results lies in the continuous assessment of the lateral variation of the Moho interface and thus well documents the Moho topography in the northern part of the profile (Figure 5a). Our results suggest that of the previously published information for the profile distance 50 to 100km, the shallower PmP Moho (Spada et al. (2013) based on Behm (2006)) is probably more accurate than the CSS profile model (Brückl et al., 2007) (see depth-enhanced Figure 5d), while in the profile distance 100 to 270km the opposite is true with the

refraction profile model showing the subhorizontal Moho at about 34km Moho. Note that while our results beyond profile distance 270km (latest at 300km) are difficult to interpret, all available CSS information calls for a distinct increase in the dip of the Moho exactly beneath the Northern Alpine Front at 300km profile distance. For further comparison, in Figure 5c, we plot on top of our GloPSI image the punctual measurements of the Moho depth obtained by depth migrated S-RF (Bianchi et al., 2014), and by the ZK analysis (Zhu and Kanamori, 2000) of P-RF (Bianchi et al., 2015) that were retrieved from stations

located within 20 km distance from EASI. We also compare our image with the Moho topography obtained by Hetényi et al. (2018) with pre-stack migration (PSM) of P-RF along EASI. Our results and the results of this latter, show good agreement from the northern end of the EASI profile to 150 km distance. In this part of the profile, the signals both from RF and GloPSI are clear (Figure 5c) and we can, therefore, infer the presence of one strong impedance contrast across the Moho. In combination with the results shown in Figure 5b we conclude the Moho is well imaged univocally by all methods in this northernmost

section. Between 150 and 270 km profile distance we notice the divergence between our GloPSI Moho image and results presented by Hetényi et al. (2018) (Figure 5c). The laterally varying differences in depth of the Moho might be caused either by errors in the crustal velocity estimates used for depth migration, or by the presence of several crustal or mantle features that deviate from being horizontally layered. Considering the Moho results of the refraction seismic profile ALP01 (Brückl et al., 2007) that are rather well resolved in this region, the latter seems unlikely. In the southernmost part of the profile (400

to 550 km distance), the steep northward dip of the Adriatic Moho interpreted by PSM imaging (Figure 5c) is not supported





by our results. As stated previously, the GloPSI method is suitable for identifying sub-horizontal to gently-dipping interfaces. In this part of the profile (440 to 550 km distance), the Moho estimates from single station analysis have been derived by depth migrated SRF (Bianchi et al., 2014). The low frequency of the used S-wave is the reason for the large errors associated with these depth estimates, moreover, from such analysis it would not be possible to separate the contribution of more than

one impedance contrast at depth. Moreover, the two different depths inferred from the same station (circled in Figure 5c) are suggesting the presence of several impedance contrasts in the crust for this section of the profile. As last comparison, previous RF studies on crustal structures (Bianchi and Bokelmann, 2014), located anisotropy at the mid-lower crust, extending from the SEMP fault southward (feature 3 in Figure 5c). From the GloPSI, in this area (SEMP and southward, lower-crust) we see a high reflectivity pattern. The co-located high reflectivity (from GloPSI) and anisotropy (from RF), are possibly due to the same

physical reasons (e.g. layering or imbrication), which contribute to fading the Moho signal beneath the Alps. In summary, the reliably resolved GloPSI results nicely complement the published results along the EASI transect derived from CSS and RF studies (Figure 5). As discussed above, the three seismic methods exhibit different strength and limitations but they are all particularly sensitive to the first-order velocity discontinuity that represents the crust-mantle boundary. The correspondence of the Moho depth obtained by the 3 different seismic methods in the northernmost 150km of the profile suggests a crust and a

Moho in this part of the northern Alpine foreland that correlates well with the models for the continental crust proposed by Mueller (1977) and Musacchio et al. (1998) and with the crustal models published for the northern foreland further W (e.g. Ye et al., 1995). Our GloPSI results and those published from CSS studies continue to correspond well further south beneath the Molasse basin (to profile distance 270km and possibly 300km) to the northern limits of the Eastern Alps. In this section of the transect, we note that the RF results show significant lateral variations in depth and also differences between the two

RF studies. Since the study of Hetényi et al. (2018) is confined to the temporary stations of the EASI profile and the study of Bianchi et al. (2015),is punctually sampling wider region including permanent stations, these differences possibly reflect lateral velocity variations in the crust beneath the Molasse basin and further south in the northernmost Alps.

In accordance with findings in the Western and Central Alps (e.g. Schmid and Kissling, 2000), our GloPSI results document a complex crustal structure beneath the Eastern Alps. While this complexity prevents us from further interpreting any signals

south of profile distance 300km, a number of studies have proposed models of the deep structure beneath the Alps. As Figures 5b and 5c show, these models differ greatly in the estimated Moho topography across the plate boundary. With the exception of the CSS longitudinal profile by Yan and Mechie (1989), all studies suffer from limitations of the method or the data set, or both, to reliably resolve the crustal structure and Moho topography in this most interesting region. Obviously, with respect to the complex velocity structure in the crust and the strongly dipping Moho interfaces that characterize the Alpine orogen, true

3D seismic methods such as, f.e., local earthquake tomography, are needed to reliably assess the 3D velocity field of the crust, subsequently allowing to correct for crustal structure when imaging the Moho topography with RF or other methods across the plate boundary.





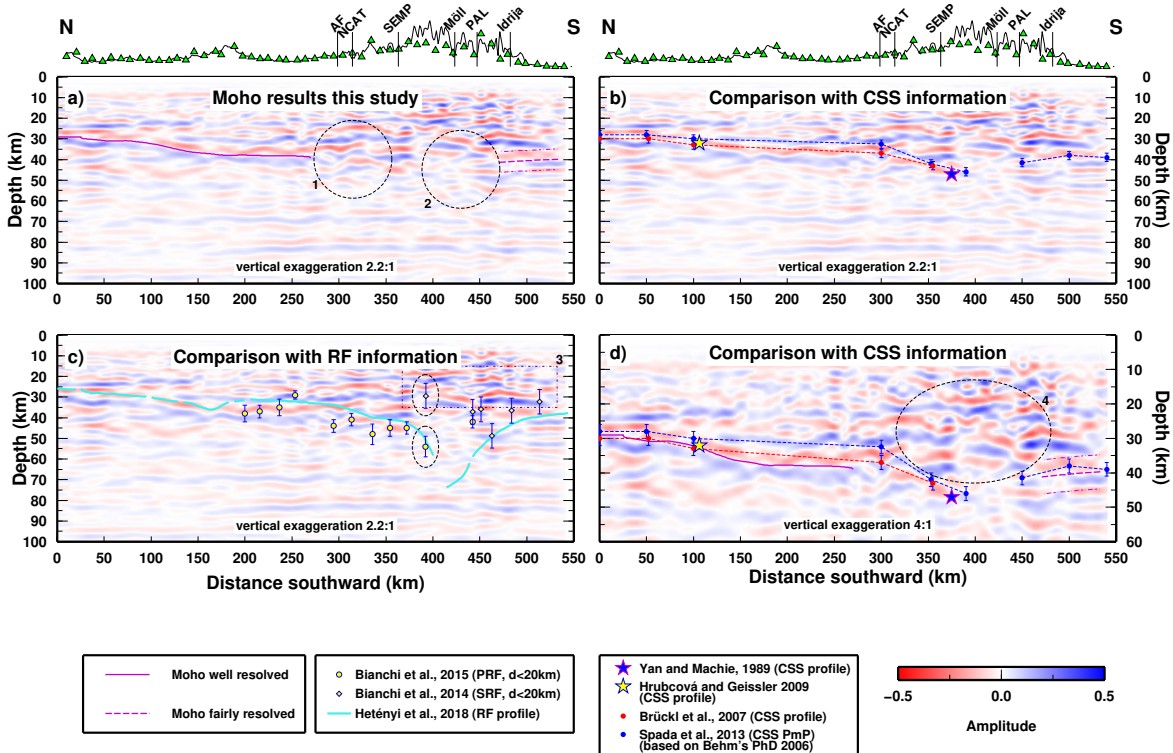

**Figure 5.** Reflectivity images of the crust and upper mantle along EASI; in the background the interpolated figure as in Figure 4b. a) Moho topography beneath the northern Alpine foreland and the Alps as detailed by results of this study. The subhorizontal and gently dipping Moho is well imaged by our global phase interferometry but the typical Moho signal disappears where the Moho steeply dips beneath the central parts of the Eastern Alps (features 1 and 2). b) Comparison with CSS information documenting the generally good correlation between our new Moho results and previous information on crustal thickness outside the Alps. c) Comparison with RF information where we evidence the co-location of the high reflectivity of crust and the detected anisotropic layer (feature 3). d) Comparison with CSS information in an enlarged version allows highlighting more detail and it reveals a nearly perfect correspondence with the PmP model (Behm, 2006; Spada et al., 2013) in the N and an equally good correspondence with the refraction seismic model (Brückl et al., 2007)) in the southern part of the foreland. The strong reverberation directly beneath the Alps (4) documents the complex internal crustal structure of the orogen.

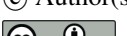



# 5   Conclusions

We applied global phase interferometry to data collected by the passive seismic deployment EASI, which crosscuts the Eastern

Alps along a 550 km long N-S profile. Inferring the crustal thickness and the nature of the Moho below the Alpine crests has been challenging in the last decades, and has led to different and often opposing interpretations. In this work, we have the opportunity to review and compare previous information on Moho depth, aside producing a new image of the crust. From north to south we can follow the different responses of the crust to the different imaging techniques (GloPSI, CSS and RF). In the northernmost part of the profile we obtain consistent depth estimates, which suggest a very simple crustal structure and a high

impedance contrast at the Moho. Between 100 and 270 km along profile, we observe diverging Moho depth estimates, which might be due to an anomalously high-velocity lowermost crustal layer, known to exist below parts of the Bohemian massif, or/and to lateral variations or local topography of the Moho interface. Between profile distance 270 and 300 km, the GloPSI does not deliver a clear image of the Moho, due to the southern dip of Europe. The segment of the profile between 300 and 550 km is the most controversial, and the one hosting the long debated and inaccessible crust-mantle boundary. The application of

this technique did not constrain the Moho topography immediately beneath the Eastern Alps, but did image the complex lower crustal structure. To univocally image the crust-mantle transition below the Eastern Alps, we further need to address this area by integrating and combining several seismic methods and by increasing the seismic station density.

*Acknowledgements.*   We thank J. Plomerová and G Hetényi for their major contribution to the realization, deployment and maintenance of the EASI seismic transect. IB thanks G. Bokelmann, F. Fuchs, P. Kolinský and the other members of the IMG Vienna, for the support given to the

realization and logistics of the Viennese contribution to the EASI project. We thank the AlpArray-EASI Field Team: Jaroslava Plomerova´, Helena Munzarova´, Ludek Vecsey, Petr Jedlicka, Josef Kotek, Irene Bianchi, Maria-Theresia Apoloner, Florian Fuchs, Patrick Ott, Ehsan Qorbani, Katalin Gribovszki, Peter Kolinsky, Peter Jordakiev, Hans Huber, Stefano Solarino, Aladino Govoni, Simone Salimbeni, Lucia Margheriti, Adriano Cavaliere, John Clinton, Roman Racine, Sacha Barman, Robert Tanner, Pascal Graf, Laura Ermert, Anne Obermann, Stefan Hiemer, Meysam Rezaeifar, Edith Korger, Ludwig Auer, Korbinian Sager, Gyo¨rgy Hete´nyi, Irene Molinari, Marcus Herrmann, Saulé

Zukauskaité, Paula Koelemeijer, Sascha Winterberg. We thank F. Bleibinhaus for providing his P-wave velocity model of the ALP2002-01 profile, used here for the depth migration. IB acknowledges the support of the Austrian Science Fund (FWF) Project J 4314-N29. We thank the SPP 4DMB project for making public the generalized tectonic map of the Alps (http://www.spp-mountainbuilding.de), which we used in Figure 1.

# 6   Data availability

The data is distributed through EIDA (European Integrated Data Archive), ETH node. Tha entire dataset is open since October 2018. The EASI network code is XT.





*Author contributions.* I.B., E.R., A.O. and E.K. contributed to the design and implementation of the research, to the analysis of the results and to the writing of the manuscript.

*Competing interests.* I.B. is part of the Topical Editor pool of this Journal.





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
