# Peer review of "Moho topography beneath the Eastern European Alps by global phase seismic interferometry"

_Solid Earth, 2020_

## Referee Comment (RC1) · Anonymous Referee #1 · 11 Dec 2020

Remarks to the Authors:

The authors of "Moho topography beneath the Eastern European Alps by global phase seismic interferometry" image the Moho depth along the transect of the EASI temporary seismic network in the Eastern Alps with GloPSI, which is a technique to extract coherent phase from the stacked auto-correlations of teleseismic waves. The authors found a simple crustal structure and hereby a clear Moho reflection phase in the northern part of the transect. While in the southern part under the higher crest of the Alps, the results indicate a more complex velocity structure with multiple strong impedance contrast in the crust and an ambiguous crust-mantle boundary. The results confirm ob-

servations in previous CSS and RF analysis. Overall, I think this manuscript is written well. However, the manuscript currently leaves room for improvement. This probably requires a minor revision. My comments consist mostly as requests for clarifications of the methodology. I hope they will be useful to transform this work into a seminal paper.

Comments and Questions:

1. The organization of the introduction section is somehow confusing, the two long paragraphs are rather tedious, making it hard to follow. I'd rather split into short paragraphs, and each short paragraph discusses just one main idea. For instance, the general introduction of 'Moho', the current research status, the main research interest, the choice of method (its advantages compared to CSS and RF) could be separate paragraphs.

2. line 34, "Anyways ...": this sentence does not seem to connect with the context.

3. line 41, "The wide-angle ...": recent seismic tomography studies give pretty reliable estimates of the Moho depth (such as Lu et al. 2018; Lu et al. 2020; Qorbani et al. 2020). I think it would be good complementary info, at least should be mentioned, in spite of the relative weak sensitivity of seismic wave traveltime to interfaces.

4. lines 85-105: could you further clarify the motivations of using GloPSI? I have difficulty in understanding why the GloPSI could provide new info beyond the RF analysis, as for instance, the influence of a complex crustal structure would affect the two methods in imaging the Moho in a similar fashion.

5. line 116, "Our ...": could you clarify the reasons for using the time range -10 to 80 s around the P-wave onset? This might involve two subquestions: i) why not use S-waves? I would also expect a clear cross-term between S and S reflection phase from auto-correlations, combined with that of P-wave, could help to interpret the final results; ii) why use a long lag time until 80 s after the P-wave onset? Does this mean that the long P-coda also contributes to the recovered cross-term between P and P

reflection phase?

6. line 124, "For ... ray parameter 0 to 0.06 s/km": I do not see the reasons for using such a ray parameter range. I roughly calculated the arrival time for different phases of the receiver side, and it seems to me that the chosen ray parameter range would not help to cancel out the "spurious arrivals", such as cross-terms between P-waves and its later reverberations (depth phases).

7. line 126, "After ...": could you clarify the choice of frequency range? I think the low-frequency content will not contribute to the final results since it is less sensitive to the interface due to finite-frequency effects. Moreover, I am afraid it will result in artifacts in the later processing procedures.

8. line 127, "spectral balancing ...": could you further explain the motivation of applying spectral balancing? This might also recap comment 7 on the usage of low-frequency content.

9. line 148, "We ...": this might be a fundamental concern: by checking fig. S1 to S8 in the supplementary materials, I have the feeling that the result highly depends on the choice of the pool of events used for imaging. In this way, the results will be more subjective and less convincing.

10. line 165: the use of "clearly visible" is somehow overrated.

11. lines 165-173: It is not clear to me the reasons behind these observations. I guess the difference between crustal features (positive, red) and mantle features (negative, blue) in the BAR image is coming from the low-frequency content in the auto-correlations, as the low-frequency representation of the reflection response. The removal of the low-frequency content leads to the change from a single impulse to blue-red-blue phase alternation in DPR image.

12. lines 180, "We also": I have difficulty in understanding the absolute values of std in Fig 3. If it is std of the amplitude, I would suggest having an additional assessment of the depth uncertainty of this cross-term between the P and P reflection phase associated with the Moho interface.

13. lines 194, "This is ...": why the SSR signals are still visible seeing that they are much less constructive than the RSR signals?

14. lines 215, "Unfortunately": I would recall comment 3, the results from recent tomographic studies.

15. lines 222, "The suggested ...": I think the GloPSI method has difficulties in imaging the Moho interface (in spite of its geometry) in the presence of a complex crustal structure (see also comment 3). As a consequence, it is hard to conclude that there exists a complex Moho topography. In other words, it is simply not imagined. This might concern the interpretation throughout the MS.

---

## Referee Comment (RC2) · Anonymous Referee #2 · 4 Jan 2021

This paper intends to apply the global phase seismic interferometry (GloPSI) method to passive seismic data acquired along the EASI profile in the Eastern Alps in order to provide additional constrains on the Moho topography in this region. First, I have to admit that I have never been fully convinced that this method is a very suitable tool to precisely image the crustal structure (compared to receiver functions for example) as it is intrinsically hard to remove artefacts like source-side reverberations (as stated by the authors and in several other published papers). Nevertheless, this is a potentially interesting contribution. However, I consider that the paper needs to be significantly improved and polished before publication. It can be appreciably shortened without losing its interest. Below I indicate, for each part of the paper, my main concerns as

[Figure]

well as other more minor comments.

Introduction

Main comment: This section is unnecessarily long and needs to be significantly shortened. It should also be better organised by clearly separating the presentation of the geodynamic context, the past studies (mostly from CSS and receiver functions) and the questions pending in the area of interest. The first part related to the Moho and Pn/Pg/PmP phases + development of CSS methods worldwide is not necessary. Later on, there are numerous back-and-forth between the presentation of the different seismic profiles, their main results in terms of Moho depths and the geodynamic implications making it very difficult to follow. The focus should be mostly on the available data and previous results in the area of interest, naming along the EASI line.

Other comments: (l19) A proper reference to Mohorovicic (1910) is lacking (l19-20) "in seismic records from intra-crustal earthquake" => lacks a verb (l27) replace "lithosphere" by "domains" (l30) CSS should englobe both refraction and reflection methods (l34) remove "for the Moho topography" (l34-35) Why "anyways"? Why unravelling the Moho beneath the Alps is a challenging task (compared to other regions)? (l45) Replace "well known" by "typical" (l48 and elsewhere) replace "W" by "west" (l46-50) The sentence is too long and should be simplified (l70) Which passive methods are you referring to? Why do you say that it is more challenging to differentiate the Moho from other interfaces from passive methods? (l71) Why such a sentence about the fact that sources and receivers are at the surface in CSS? (l71-79) This discussion on migration and 3D effects is unnecessary long, unclear, and I don't see the link with the present study. (l80-81) Where should be this "Moho triple junction" or "Moho gap"? (l86) Define the acronym "RF" used for receiver functions (l90) "The Moho is not imaged": Add an adjective like "well" (l91-92) "inconclusive converted signals by RF" does not mean anything (l95-96) I don't understand what you mean by "turning passive measurements" (namely earthquake signals) "into deterministic seismic responses". (l96) Which "principle" are you talking about? (l94-98) Unnecessary long (Figure 2): What are the units

of the color scales? + Add orientation on top of Fig 2A (and Fig 3A)

Data and method

Main comment: The presentation of the GloPSI technique should be improved / simplified / better organised. On the one hand, there are unnecessary repetition making it difficult to follow and understand the various processing steps. On the other hand, it lacks more precise information on these various processing steps, how they are implemented and their respective role. For example, the author should remind what is the muting of the delta-pulse and why it is required. Same comment applies for the multiple suppression. Based on Figure 2b and 2c it is unclear to me what is the influence of this processing step (I just barely only see a reduction of the amplitudes between Fig 2b and 2c). Also, the values chosen at each step should be given (for example which filter is applied to remove the "delta pulse") and the effect of modifying these values on the resulting images can be discussed. For example, it would be interesting to test/show the effect of spectral balancing and be more precise about the way it has been implemented (for reproductivity of the study).

Other comments: (l113-114) Unnecessary repetition of the fact that you also use direct P waves (l114) Be more precise about the distance range around 150° you exclude (l115) Replace P by PKP (l116) What do you mean by "The 64 events display a high station coverage"? (rephrase) (l121) replace "result" by "part" + explain what you mean by "muting the delta pulse" (the explanation appears later in the text but should be improved) (l123) There is no moveout correction performed before the stacking? (l124) What are the "spurious phases created"? (probably refers to SSR) (l127-128) The explanation for the "spectral balancing" is unclear. To me the objective is to get closer to the spectrum of a delta-like function (l130) If the autocorrelation is only applied on the phase spectrum then the spectral balancing (which is performed on the amplitude spectrum I imagine) is unnecessary (l134) Be more precise about the static correction you applied and explain the technique used to eliminate the surface-related multiples (l139-140) You already presented the Alp01 profile in the introduction

Results

Main comments: - The selection of the 27 events out of the 64 available ones is still unclear to me. First, I don't see major differences between Figure 2 (with 27 events) and figure S8 (with all the 64 events). Secondly, the authors states in the text (l193-l197), supported by the interpolated reflectivity images (Fig. 4a and 4b), that using the 64 events tends to reduce source-side reverberations in the reflectivity images. But later they favour the results based on the selected 27 events (l201 : "Consequently, we decide to use image obtained with 27 events . . ."). Third, it is unclear to me why some source-side reverberations (SSR) should cancel out in one part of the profile (northern part) and not on the other part (southern part). Is there a physical reason for that?

- Crust/Moho signature: I don't understand why in the BAR images the crust should (physically) correspond to positive (red) features and the mantle to negative (blue) ones. Is it an effect of the high pass filter applied to the initial data? Why (physically) the Moho would appear as a blue-red-blue triplet after muting the delta pulse? Moreover, such triplet is not always well seen (or with a symmetric shape) like for stations ∼15 to 25 on Fig. 2.

- Phase at 12s for the southern stations (l177-180): I don't understand the argument bring by the author to consider this phase as an artefact for the southern stations but not for the northern stations (this phase is also seen for stations ∼17 to ∼30). If it is a source-side reverberation it should appear for all the stations (except if the authors selected some events only for the southern stations . . . which should be avoided).

- Standard deviation (Fig. 3 and S9): It is difficult to compare both Fig. 3 and S9 as the scale is different. Personally, I don't see a major difference between the images . . . Moreover, the higher std's are observed for time ranges where Moho reflected phases are expected. Therefore, can we really interpret the reflectivity images in this time range? (the authors states at lines 185-186: "The time location of larger standard deviation [. . .] should not be interpreted geometrically" ! )

- Finally, the authors say that "it is quite likely that the entire [...] signal below the Alps is dominated by artefacts" (l200-201) but later they often interpret several features (Moho, intra-crustal structure) in this area (cf. l206, l215, l222-223 + Discussion part and Figure 5).

Other comments: (l144) The Fresnel extension should depend on the frequency used and the depth (l145-147) Unclear sentence. Please rephrase (l146) Why do you say that "only the Moho fulfills these requirements"? What about a continuous intra-crustal or upper-mantle reflector? (l156) The fact that the authors have "more" phases available than Ruigrok and Wapenaar (2012) is due to different selection criteria (Ruigrok and Wapenaar used only M>6 events and PKiKP and PKIKP) (l162) In Figures 2a, S7a, S8a (BAR images), the Moho rather corresponds to the limit between positive signals and negative signals (although I don't really understand why) rather than "a strong positive signal". (l165-169) Explain the reason why the Moho should correspond to a blue-red-blue feature + avoid the repetitions among the various sentences (l169-l171) Give a physical reason why the crustal "features" should be positive (red) and the mantle "features" should be negative (blue) in the BAR images. (l187-l191) Various part of this paragraph are unclear ("source of wave energy are the selected earthquake", "the subset 27 events that are closer clustered in space", "the second source denotes the targeted structure", ...) and should be rephrased. (l204) Why do the authors choose to "pick the lower zero crossing (within the blue-red-blue triplet)" as the Moho and not the central positive pick?

(Figure 4) what type of interpolation is used? What is the unit of the amplitude color scale? Why is it 5 times higher than on figure 2?

Discussion and conclusion

Main comments: - The authors compare they Moho depth estimates to other studies and challenge these previous results (especially the ones from Hetenyi et al. (2018)). But how sensitive is their migrated image (and corresponding estimation of Moho depth

[Figure]

/ topography) to the uncertainties on the velocity model they use for the migration? In the conclusion they mention a potential anomalously high-velocity lowermost crust beneath the Bohemian massif but it is unclear to me if they have it (or Hetenyi et al. (2018)) in their velocity model and what would be the effect to include/remove it.

- Both in the introduction and the conclusion the authors mention "opposing" views and geodynamic interpretations of the seismic profile in the area in the literature. In this section it would be good to better indicate which of these previous views are supported (and which ones are not) by the results of their study.

Other comments: (l260) Change "not supported" by "not seen" (l280-ll282) Based on this sentence it is still unclear to me why the GloPSI results differ from the RF results from Hetenyi et al. (2018) between 150 and 300km. Both are based on the same EASI stations. Do you mean that the "lateral velocity variations in the crust" (l282) are not properly taken into account in Hetenyi et al. (2018)'s velocity model used in their migration (as stated before l256-l257)? (l303) replace "Europe" by "European plate". (Figure 5) Legend of Figure 5 indicates that "the Moho signal disappears where the Moho steeply dips beneath the central part of Eastern Alps" => But if the Moho disappears you cannot say that it is steeply dipping!! Please rephrase.

---

## Author Comment (AC1) · 13 Feb 2021

We thank the reviewer for the constructive comments. Here we list point by point the Reviewer's comments (*in italic*) and our reply.

*R#1*

*1. The organization of the introduction section is somehow confusing, the two long paragraphs are rather tedious, making it hard to follow. I'd rather split into short paragraphs, and each short paragraph discusses just one main idea. For instance, the general introduction of 'Moho', the current research status, the main research interest, the choice of method (its advantages compared to CSS and RF) could be separate paragraphs.*

-- We have shortened the introduction, and split the text according to the topics, as suggested by both reviewers.

*2. line 34, "Anyways ...": this sentence does not seem to connect with the context.*

-- We modified the text to connect this sentence with the surrounding text.

*3. line 41, "The wide-angle ...": recent seismic tomography studies give pretty reliable estimates of the Moho depth (such as Lu et al. 2018; Lu et al. 2020; Qorbani et al. 2020). I think it would be good complementary info, at least should be mentioned, in spite of the relative weak sensitivity of seismic wave traveltime to interfaces.*

-- We thank the reviewer for the suggestion, and added the references in the introduction, together with one more reference from Molinari et al "3D crustal structure of the Eastern Alpine region from ambient noise tomography" Results in Geophysical Sciences, 2020. Anyways the ambient noise tomography used in these works is not a good tool for inferring the presence of impedance contrasts at depth (as the RF and GloPSI are), rather it is good for identifying lateral variations.

*4. lines 85-105: could you further clarify the motivations of using GloPSI? I have difficulty in understanding why the GloPSI could provide new info beyond the RF analysis, as for instance, the influence of a complex crustal structure would affect the two methods in imaging the Moho in a similar fashion.*

-- In this work our intention is to both: 1-provide new and additional information on the depth of the Moho, and 2- test which similarities and difference the two techniques (RF and GloPSI) retain. Both of them use transmitted waves and are sensitive to the presence of changes in acoustic impedance at depth. There is an important difference: RF only show something when the waves make some angle of incidence with respect to the reflector. GloPSI, on the other hand, only retrieves a reflection when the angles of incidence with respect to the interface are close to zero. Hence, some reflectors would be seen with GloPSI and not with RF and vice versa.
And as we show in the end in this work, both the techniques have poorer resolution when the crustal structure is complex. We thank the reviewer for this point, and we add this in the introduction [lines 105-109]

*5. line 116, "Our ...": could you clarify the reasons for using the time range -10 to 80 s around the P-wave onset? This might involve two subquestions: i) why not use S-waves? I would also expect a clear cross-term between S and S reflection phase from auto-correlations, combined with that of P-wave, could help to interpret the final results; ii) why use a long lag time until 80 s after the P-wave onset? Does this mean that the long P-coda also contributes to the recovered cross-term between P and P reflection phase?*

--A sentence has been added to clarify that we want to include all the (receiver-side) scattering following the direct P-wave. For most events, until about 80 seconds still reverberations come in, which all contribute to the receiver-side illumination.
Mixing P and S waves is not desired as an P-wave velocity model is later used for migration.
An S-wave implementation of GloPSI is discussed in Frank et al. (2014). Constructing an S-wave reflectivity image would also be useful for EASI, e.g., to image possible presence of melt. However, this is outside of the scope of the current manuscript.

*6. line 124, "For ... ray parameter 0 to 0.06 s/km": I do not see the reasons for using such a ray parameter range. I roughly calculated the arrival time for different phases of the receiver side, and it seems to me that the chosen ray parameter range would not help to cancel out the "spurious arrivals", such as cross-terms between P-waves and its later reverberations (depth phases).*

--You are correct, the ray parameter range determines the aperture of the virtual source that is constructed. It does not say anything about spurious terms being stacked out or not. In the same section, we add another sentence to clarify that stacking over events with different depths is needed in order to suppress spurious cross terms due depth phases.

*7. line 126, "After ...": could you clarify the choice of frequency range? I think the low frequency content will not contribute to the final results since it is less sensitive to the interface due to finite-frequency effects. Moreover, I am afraid it will result in artefacts in the later processing procedures.*

-- You are right here. This is an important detail we did not describe well. Additional text has now been added in the Method section. Before autocorrelation we retain a quite wide frequency band (0.04-0.8 Hz). The lower part of this frequency band has poor signal-to-noise ratio and a limited information content on the receiver-side structure. However, these lower frequencies are still effective in obtaining a sharper delta pulse at t=0 after autocorrelation. We can then use a shorter time window to remove the delta pulse and therewith we retain a larger part of the reflections. These lower frequencies are subsequently removed with a high-pass filter with a cut-off frequency at 0.2 Hz.

*8. line 127, "spectral balancing ...": could you further explain the motivation of applying spectral balancing? This might also recap comment 7 on the usage of low-frequency content.*

-- We added additional explanation on this point. Seismic interferometry is the evaluation of an integral equation. Each contribution in the integrand should have a similar frequency content. If not, the stationary-phase process of enhancing reflections at the physical travel time, fails. Individual earthquakes have largely varying spectral content due to different source properties (corner frequencies) and different propagation effects (elastic and anelastic attenuation). Hence, some spectral balancing needs to be applied prior to seismic interferometry.

*9. line 148, "We ...": this might be a fundamental concern: by checking fig. S1 to S8 in the supplementary materials, I have the feeling that the result highly depends on the choice of the pool of events used for imaging. In this way, the results will be more subjective and less convincing.*
*[now line 172]*

-- Indeed the results do depend on the pool of events used, as in any other technique in geophysics. With showing the figure S1 to S8 we wanted to point out the outcomes when using an unbalanced number of events from the two sides (from north and south in this case) of the transect. When we choose events mainly from the south or from the north as in figure S4 and S5, there is no ray crossing beneath the profile, and the spurious phases (generated from the same backazimuthal directions) are not cancelling, rather summing up.
With a high number of events, balanced from both sides of the transect, the receiver side-signals are stacking "positively" and the real features are emerging, as in figures 2 and 4 in the main text.
Since we got this remark from R1, we understand that these concepts are not clear in the main text, and therefore modify the text in lines 174-175 and 184-175.

*10. line 165: the use of "clearly visible" is somehow overrated.*

-- We understand this remark and change *"clearly visible"* with "visible as blue-red-blue triplet"

*11. lines 165-173: It is not clear to me the reasons behind these observations. I guess the difference between crustal features (positive, red) and mantle features (negative, blue) in the BAR image is coming from the low-frequency content in the autocorrelations, as the low-frequency representation of the reflection response. The removal of the low-frequency content leads to the change from a single impulse to bluered- blue phase alternation in DPR image.*
[now lines

--We have removed this observations, since they are confusing the reader and since the interpretation needs to be done on the migrated image only

*12. lines 180, "We also": I have difficulty in understanding the absolute values of std in Fig 3. If it is std of the amplitude, I would suggest having an additional assessment of the depth uncertainty of this cross-term between the P and P reflection phase associated with the Moho interface.*

--The standard deviation is now expressed in % with respect to the maximum amplitudes in the respective panel in figure 2.
--We have added in "Table 2" the maximum and minimum Moho depths estimated by the bootstrapped images, for the distances between 0 and 270 km along profile, that correspond to the reliable Moho depths inferred by this study.

*13. lines 194, "This is ...": why the SSR signals are still visible seeing that they are much less constructive than the RSR signals?*

--We rewrote the part on SSR, as also replying to the comments of R2.3. There is no physical reason why some SSR should cancel out in the northern part when stacking in 64 events instead of 27 and not in the southern part of the profile.

*14. lines 215, "Unfortunately": I would recall comment 3, the results from recent tomographic studies.*

--We have added some references to those studies too

*15. lines 222, "The suggested ...": I think the GloPSI method has difficulties in imaging the Moho interface (in spite of its geometry) in the presence of a complex crustal structure (see also comment 3). As a consequence, it is hard to conclude that there exists a complex Moho topography. In other words, it is simply not imagined. This might concern the interpretation throughout the MS.*

We have modified the text by excluding the Moho topography and referring to the internal crustal structure.

---

## Author Comment (AC2) · 13 Feb 2021

We thank the reviewer for the constructive comments. Here we list point by point the Reviewer's comments (*in italic*) and our reply.

R#2

*Introduction*
*R2.1 Main comment: This section is unnecessarily long and needs to be significantly shortened.*
*It should also be better organised by clearly separating the presentation of the geodynamic context, the past studies (mostly from CSS and receiver functions) and the questions pending in the area of interest. The first part related to the Moho and Pn/Pg/PmP phases + development of CSS methods worldwide is not necessary. Later on, there are numerous back-and-forth between the presentation of the different seismic profiles, their main results in terms of Moho depths and the geodynamic implications making it very difficult to follow. The focus should be mostly on the available data and previous results in the area of interest, naming along the EASI line.*

--The text of the introduction is now re-arranged by separating the geodynamics parts, which are now in the initial paragraphs, and the previous studies, which are following. The digression about the methods and their resolution is cut out, and is partially moved to the Discussion section.

*Other comments:*

*(l19) A proper reference to Mohorovicic (1910) is lacking*
--Added

*(l19-20) "in seismic records from intra-crustal earthquake" => lacks a verb*
 --we modified with "produces in seismic records"

*(l27) replace "lithosphere" by "domains"*
--replaced

*(l30) CSS should englobe both refraction and reflection methods*
--we cancelled the repetition

*(l34) remove "for the Moho topography"*
--removed

*(l34-35) Why "anyways"? Why unravelling the Moho beneath the Alps is a challenging task (compared to other regions)?*
--This sentence is deleted

*(l45) Replace "well known" by "typical"*
--replaced

*(l48 and elsewhere) replace "W" by "west"*
--replaced  (and elsewhere too)

*(l46-50)  The sentence is too long and should be simplified*
--This sentence has been cut out

*(l70) Which passive methods are you referring to? Why do you say that it is more challenging to differentiate the Moho from other interfaces from passive methods?*
--This sentence has been cut out

*(l71) Why such a sentence about the fact that sources and receivers are at the surface in CSS?*
--We cancel this sentence since it is unnecessary

*(l71-79) This discussion on migration and 3D effects is unnecessary long, unclear, and I don't see the link with the present study.*
--We cancel this sentence from the introduction

*(l80-81) Where should be this "Moho triple junction" or "Moho gap"?*
--We mark this in Figure 1

*(l86) Define the acronym "RF" used for receiver functions*
--Defined

*(l90) "The Moho is not imaged": Add an adjective like "well"*
--Added

*(l91-92) "inconclusive converted signals by RF" does not mean anything*
--changed by weak

*(l95-96) I don't understand what you mean by "turning passive measurements" (namely earthquake signals) "into deterministic seismic responses".*
--The sentence is cancelled

*(l96) Which "principle" are you talking about?*
--The principle of generating new seismic responses by virtual sources; but the sentence is deleted now from the text.

*(l94-98) Unnecessary long*
--we have shortened by deleting the sentences outlined above.

*(Figure 2): What are the units of the color scales? + Add orientation on top of Fig 2A (and Fig 3A)*
-- The amplitude is adimensional; the raw signals are generated by the autocorrelation of the earthquakes and the final signal for each station is retrieved by stacking all autocorrelation functions.
--We added the orientation on fig 2A and 3A

*Data and method*
*R2.2 Main comment: The presentation of the GloPSI technique should be improved / simplified / better organised. On the one hand, there are unnecessary repetition making it difficult to follow and understand the various processing steps. On the other hand, it lacks more precise information on these various processing steps, how they are implemented and their respective role. For example, the author should remind what is the muting of the delta-pulse and why it is required. Same comment applies for the multiple suppression. Based on Figure 2b and 2c it is unclear to me what is the influence of this processing step (I just barely only see a reduction of the amplitudes between Fig 2b and 2c). Also, the values chosen at each step should be given (for example which filter is applied to remove the "delta pulse") and the effect of modifying these values on the resulting images can be discussed. For example, it would be interesting to test/show the effect of spectral balancing and be more precise about the way it has been implemented (for reproductivity of the study).*

*Other comments:*
*(l113-114) Unnecessary repetition of the fact that you also use direct P waves*
*(l114) Be more precise about the distance range around 150 you exclude*
*(l115) Replace P by PKP*
*(l116) What do you mean by "The 64 events display a high station coverage"? (rephrase)*
*(l121) replace "result" by "part" + explain what you mean by "muting the delta pulse" (the explanation appears later in the text but should be improved)*
*(l123) There is no moveout correction performed before the stacking?*
*(l124) What are the "spurious phases created"? (probably refers to SSR)*
*(l127-128) The explanation for the "spectral balancing" is unclear. To me the objective is to get closer to the spectrum of a delta-like function*
*(l130) If the autocorrelation is only applied on the phase spectrum then the spectral balancing (which is performed on the amplitude spectrum I imagine) is unnecessary*
*(l134) Be more precise about the static correction you applied and explain the technique used to eliminate the surface-related multiples*

--Thanks for your detailed suggestions. We largely rewrote this section, adding explanation and details on the processing to allow reproducibility. Also we improved the structure to make it easier to read.

*(l139-140) You already presented the Alp01 profile in the introduction*
--we have cancelled the description of the profile

*Results*
*Main comments:*
*R2.3 - The selection of the 27 events out of the 64 available ones is still unclear to me. First, I don't see major differences between Figure 2 (with 27 events) and figure S8 (with all the 64 events). Secondly, the authors states in the text (l193- l197), supported by the interpolated reflectivity images (Fig. 4a and 4b), that using the 64 events tends to reduce source-side reverberations in the reflectivity images. But later they favour the results based on the selected 27 events (l201 : "Consequently, we decide to use image obtained with 27 events : : :"). Third, it is unclear to me why some source-side reverberations (SSR) should cancel out in one part of the profile (northern part) and not on the other part (southern part). Is there a physical reason for that?*

--The fact that there are not large differences between using a pool of 27 or 64 events is a good result. It means that the pool of events chosen is well balanced, both the images are equivalent in terms of results. We have now decided to use the image produced by the pool of 64 events, where the suppression of SSRs is better.
We rewrote the part on SSR. There is indeed no physical reason why some SSR should cancel out in the northern part when stacking in 64 events instead of 27 and not in the southern part of the profile.

*R2.4 - Crust/Moho signature: I don't understand why in the BAR images the crust should (physically) correspond to positive (red) features and the mantle to negative (blue) ones. Is it an effect of the high pass filter applied to the initial data? Why (physically) the Moho would appear as a blue-red-blue triplet after muting the delta pulse? Moreover, such triplet is not always well seen (or with a symmetric shape) like for stations 15 to 25 on Fig. 2.*

--Good point. We cannot think of a sensible physical explanation. In the BAR images, the crust appears to correspond to the red part. In the northern part this holds true. However, further to the south where the crust is thicker, it does not hold true. The response at the very low frequencies is largely spurious. We have added the explanation of why we first include these frequencies (to obtain a narrow delta pulse that can be removed without muting much of the reflections response) and later remove them again (because these low frequencies have very limited information content on the crust).
Only after removing the low frequencies, the actual reflections are visible without low-frequency interference. Thus, also the Moho is shown as a positive pulse (red) with small sidelobes (blue).

*R2.5 - Phase at 12s for the southern stations (l177-180): I don't understand the argument bring by the author to consider this phase as an artefact for the southern stations but not for the northern stations (this phase is also seen for stations 17 to 30). If it is a source-side reverberation it should appear for all the stations (except if the authors selected some events only for the southern stations : : : which should be avoided).*

--We agree with you and remove this text, and we stated that only the Moho is interpreted.

*R2.6 - Standard deviation (Fig. 3 and S9): It is difficult to compare both Fig. 3 and S9 as the scale is different. Personally, I don't see a major difference between the images : : : Moreover, the higher std's are observed for time ranges where Moho reflected phases are expected. Therefore, can we really interpret the reflectivity images in this time range? (the authors states at lines 185-186: "The time location of larger standard deviation [: :] should not be interpreted geometrically" ! )*

--As there is little difference between figure 2 and S8, there is little difference between figure 3 and S9. We have used the colorscale that maximises the differences in each single plot. The standard deviation is now expressed in % with respect to the maximum amplitudes in the respective panel in figure 2 and S8. The main difference to be noted in these figures is between the northern and southern parts of the profile. The northern part has lower std than the southern part. The northern part has relative higher std for time arrivals (~8 s) earlier than the times arrivals of the moho signal (about 10 to 12 s). This reinforces our observations that the constrains that we give to the southern part of the profile are poor. We have modified the text.

*R2.7 - Finally, the authors say that "it is quite likely that the entire [: : :] signal below the Alps is dominated by artefacts" (l200-201) but later they often interpret several features (Moho, intra-crustal structure) in this area (cf. l206, l215, l222-223 + Discussion part and Figure 5).*

We agree that this was an awkward statement. Also below the Alps, a large part of the imaged features is likely real. However, the further in depth one goes, the bigger the chance that imaged reflectivity is spurious, due to imaging remaining complex reverberations instead of primary reflections. These complex reverberations are more likely to exist in the Alps than at other parts of the transect.
We rewrote this part of the text and are now more specific on what could be possible artifacts.

*Other comments:*
*(l144) The Fresnel extension should depend on the frequency used and the depth*
--frequency and depth are added

*(l145-147) Unclear sentence. Please rephrase*
--it is rephrased

*(l146) Why do you say that "only the Moho fulfills these requirements"? What about a continuous intra-crustal or upper-mantle reflector?*
--We wanted to highlight that the Moho is the strongest first order signature, and it is a global feature [while other reflectors (in the crust or upper mantle) might have a local or regional nature]. Since the sentence seems to be confusing, we rephrase it.

*(l156) The fact that the authors have "more" phases available than Ruigrok and Wapenaar (2012) is due to different selection criteria (RuigrokandWapenaar used only M>6 events and PKiKP and PKIKP)*

--Indeed, this is what we want to highlight. The interferometry in Ruigrok and Wapenaar could give a solid image with 17 earthquakes only, therefore our application (in both examples 27 and 64) delivers a solid image

*(l162) In Figures 2a, S7a, S8a (BAR images), the Moho rather corresponds to the limit between positive signals and negative signals (although I don't really understand why) rather than "a strong positive signal".*
*(l165-169) Explain the reason why the Moho should correspond to a blue-red-blue feature + avoid the repetitions among the various sentences*
*(l169-l171)Give a physical reason why the crustal "features" should be positive (red) and the mantle "features" should be negative (blue) in the BAR images.*

--We have addressed these three remarks already in R1.11 and R2.4

*R2.8 (l187-l191) Various part of this paragraph are unclear ("source of wave energy are the selected earthquake", "the subset 27 events that are closer clustered in space", "the second source denotes the targeted structure", : : :) and should be rephrased.*
--We have rephrased this

*(l204) Why do the authors choose to "pick the lower zero crossing (within the blue-red-blue triplet)" as the Moho and not the central positive pick?*
--We have chosen it  simply because it is clearer to see, the error introduced in the Moho depth by picking the zero-crossing or the max amplitude of the red phase is within the errors given by the migration with a velocity model rather than another one. Anyways, in order to be in line with previous studies we now mark the max amplitude of the positive signal.

*(Figure 4) what type of interpolation is used? What is the unit of the amplitude color scale? Why is it 5 times higher than on figure 2?*
*--A bilinear interpolation has been done. We substitute the colorscale with a normalized colorscale.*

*Discussion and conclusion*
*R2.9 Main comments: - The authors compare they Moho depth estimates to other studies and challenge these previous results (especially the ones from Hetenyi et al. (2018)). But how sensitive is their migrated image (and corresponding estimation of Moho depth / topography) to the uncertainties on the velocity model they use for the migration? In the conclusion, they mention a potential anomalously high-velocity lowermost crust beneath the Bohemian massif but it is unclear to me if they have it (or Hetenyi et al. (2018)) in their velocity model and what would be the effect to include/remove it.*

--Both this study and Hetenyi et al.  (2018) are not including for the depth migration the high velocity layer regionally identified by Hrubcova et al (2005). Hetenyi uses the velocity models shown in Figure S10 (EP crust from Molinari and Morelli 2011). We have added in the discussion some considerations about including this layer in the depth migration.

*R2.10 - Both in the introduction and the conclusion the authors mention "opposing" views and geodynamic interpretations of the seismic profile in the area in the literature. In this section it would be good to better indicate which of these previous views are supported (and which ones are not) by the results of their study.*

--Unfortunately, the place where the previous Moho estimates are differing the most is where our interferometric image is losing resolution, therefore we cannot support one or another view. All previous models are very precious in order to shed light on the study area, and are adding complementary information about the structures at depth, and we conclude that only a comprehensive 3D reconstruction might shed light on the crust-mantle boundary in the area.

*Other comments:*
*(l260) Change "not supported" by "not seen"*
*--changed*

*(l280-ll282) Based on this sentence it is still unclear to me why the GloPSI results differ from the RF results from Hetenyi et al. (2018) between 150 and 300km. Both are based on the same EASI stations. Do you mean that the "lateral velocity variations in the crust" (l282) are not properly taken into account in Hetenyi et al. (2018)'s velocity model used in their migration (as stated before l256-l257)?  (l303) replace "Europe" by "European plate".*
-- Hetenyi et al (2018) uses the regional Vp and Vs model from Molinari and Morelli (2011) for migrating the Moho Ps conversions. We instead, use a  transect specific Vp model (Brueckl et al, 2007) which is much more detailed and reliable for the EASI transect.  The RF image of Hetenyi et al (2018)  is based on velocity models with a much poorer resolution and the Vs model has much higher uncertainty than the Vp model that we could take advantage of. We have added this sentence in the text.

*(Figure 5) Legend of Figure 5 indicates that "the Moho signal disappears where the Moho steeply dips beneath the central part of Eastern Alps" => But if the Moho disappears you cannot say that it is steeply dipping!! Please rephrase.*
--Rephrased

---

## Editor Decision (ED1)

Comments on the revised version of ms. se-2020-179 "Moho topography beneath the Eastern European Alps by global phase seismic interferometry" by I. Bianchi et al.

The manuscript still needs minor corrections before it can be accepted for publication. Although some effort has been made by the authors to reduce the length of the introduction in the revised version, there is still a need to improve its organization and reduce its length by focusing on the actual objectives of the study. I also suggest to update the discussion/conclusion by commenting on the results of a recent study that was published recently, and that uses the data of the densest broadband seismic array operated in the Alpine region, the Swath-D experiment (Sadeghi-Bagherabadi et al., 2021).

More detailed comments are listed below.

1) Abstract, l. 7: you write that your method "well images the topography of the Moho in regions where it shows a nearly planar behaviour… from the Bohemian massif to beneath the Northern Calcareous Alps". Is this really because the Moho topography is planar, or because it is reflective (corresponds to a strong velocity contrast)?

2) Abstract, l. 9: what is a "typical" crust-mantle boundary?

3) Abstract, l. 10: "absence of an Adriatic crust made of laterally continuous layers smoothly descending southwards". So, what is present? Is it a "structurally complex and faulted internal crustal structure" as suggested in the next sentence (but for the Alpine crust)?

4) Abstract, l. 11: why do you conclude on a "structurally complex and faulted internal Alpine crustal structure". This contradicts the earlier sentence when you write that the Moho of the Northern Calcareous Alps is clear. They are not part of the Alps?

5) Introduction, l. 16-18: useless details in sentence "After the closure of major and minor oceans, … continental parts of the much smaller plate Adria collided". Could be summarized to "After the closure of the Alpine Tethys, the European continental margin collided with the small Adria plate".

6) Introduction: in fact, the previous comment is one example of an unnecessarily long and detailed sentence, and there are quite a few like that in the introduction. I am not sure that it is useful to cite all the experiments that have produced geophysical images in that region. You should focus the introduction on the key question that you address in that paper, which is the apparent Moho gap of Spada et al. (2013).

7) Introduction, l. 21-22: is there really "a general agreement that the European and the Adriatic Moho are offset across the plate boundary in the Alps"? Which publications state that?

8) Introduction, l. 23-44: I would suggest to avoid listing here all CSS experiments in the Alps and to keep only those related to the Eastern Alps.

9) Introduction, l. 45-47: you write that most information about the Moho is derived from CSS experiments but you refer to publications such as Diehl et al. (2009) that only deals with earthquake sources. There is no contradiction, but this reference is inaccurate in this context.

10) Introduction, l. 48: you should explain what the "Moho triple junction" of Brückl et al. is, because it is probably one of the questions that you want to address.

11) Figure 1: you use similar thin plain lines (of different colors) to show very different features such as tectonic structures, the triple junction of Brückl et al. that refers to

the Moho structure and to outline the area of Moho gap by Spada et al. This makes the figure confusing. I would suggest using different types of lines, following geological standards for the Alpine front for example and a filled polygon for the Moho gap area.

12) Introduction, l. 62-64: the sentence on recent ambient-noise tomography studies brings no useful information. I guess you mean that these ANT studies are more valuable for imaging velocity heterogeneities than imaging Moho depth variations. This is right, but it should be better explained. Moreover, some of the works you cite don't even reach Moho depth while others do and provide clues on the topography of velocity contours used as proxies for the Moho. This is worth mentioning.

13) Introduction, l. 69: rephrase unclear sentence ".. and stacking primarily global phases; waves that travel across the core…".

14) Introduction, l. 74: what do you mean by "considerably greater than zero"?

15) Introduction, l. 75: correct "Alpine reflectively". Do you mean reflectivity of structures of the Alpine crust?

16) Introduction, l. 75-76: sentence "In other…2019)" is out of context.

17) Section 2.1, l. 88-89: did you discard entire event recordings or did you only discard time windows with multiple phases? Please rephrase.

18) Section 2.2, l. 97: please rephrase "selecting minus the causal result and muting the delta pulse".

19) Section 2.2, l. 106: I guess "rupture effects" means "earthquake source effects".

20) Section 2.2, l. 112: by "reflectivity from the lithosphere at the source", you probably mean "spurious signals from the lithospheric structure at the source side".

21) Section 2.2, l. 114: step without "s"

22) Results, l. 161: replace "especially receiver-side reflectivity is shown on these images" by "these images mostly show receiver-side reflectivity"

23) Results, l. 171: you write that you decide "to focus (your) interpretation on the Moho topography in the northern part of the profile". This is surprising at this step of the paper because the most interesting objective is the "Moho gap" in the southern end. Do you mean that you quickly give up on bringing in new constraints on the most interesting southern part, and that you will not discuss this part further?

24) Results, l. 182: by "suggest the signals representing at least in parts internal crustal structure", do you mean that the amplitude difference between signals at crustal depth in the northern and southern parts suggests that at least part of the signals in the south side can be attributed to actual crustal structure?

25) Results, l. 182-185: The sentence "Unfortunately, the 3D crustal structure of the Eastern Alps below 15 km depth is still poorly known … with reference to the tectonic style and geologic evolution of the orogeny (e.g. Willingshofer et al., 2013; Rosenberg and Kissling, 2013, and references therein)" is too long and unclear, and it is partly wrong. I would consider that the crustal structure of the Eastern Alps, with TRANSALP and EASI, has been studied by as many tomography studies as the Western Alps with the CIFALPS profiles and ECORS-CROP. The crustal structure of the Central Alps is more poorly know since it has only been studied by the NFP-20 deep-seismic sounding profiles, and no dense passive seismic experiment. The reference that you give (Kissling et al., 2006) presents a synthesis of what was known at the time of writing, that is before a number of recent experiments in the Western and Eastern Alps, including EASI. You should update your reference list. I don't know

Behm et al. (2006) which is an unpublished PhD thesis whose citation is useless. You also cite Lu et al. (2020) that covers the entire Alps, and not only the Western and Central Alps, and provides the Vs structure at depth >15 km in contradiction with your sentence. Qorbani et al. (2020) does cover only the Eastern Alps to ~40 km depth, also in contradiction with your sentence. Molinari et al. (2020) and Sadeghi-Bagherabadi et al. (2021) also focus on the crustal structure of the Eastern Alps. That's a lot of publications on the crustal structure of the E-Alps in the end! The problem of the lack of clear images of the structure of the lower crust and Moho beneath the Tauern window is obviously not due to the lack of data. I don't understand what you mean by "and with reference to the tectonic style and geologic evolution of the orogeny". Please clarify.

26) Results, l. 185-186: In the next sentence, you write that you expect a complex crustal structure and you cite a review paper (Handy et al., 2015) that deals with palinspatic reconstructions and slab geometry. Again, a tomography paper that shows that imaging the lower crust is particularly difficult beneath the Tauern window, like Hetenyi et al. (2018) is more adequate. You should maybe erase these 2 sentences and leave only the one of l. 187-189, which is much more correct and accurate.

27) Discussion, l. 207-210: When you write "the strength of (your) new results lies in the continuous assessment of the lateral variation of the Moho interface… in the northern part of the profile", you seem to forget the RF results of Hetenyi et al. (2018) who were the first to provide a continuous image of the depth variations of the Moho beneath the same profile. This is surprising as the first author of the present paper is a co-author of Hetenyi et al. (2018). You should start the discussion by comparing with their results. This sentence is also contradictory with the one of l. 224 "we conclude the Moho is well imaged univocally by all methods in this northernmost section". If all methods work well in that part of the profile, imaging the same Moho as others cannot be the strength of your new results.

28) Discussion, l. 210: You cannot tell that the Moho model of Spada et al. is more accurate than the one by Brückl et al. only because the first one better fits your Moho depth estimate. The three Moho depth models depend on the velocity models used to convert time to depth. You use the Vp model by Brückl et al. shown in Fig. S10. I would therefore expect your Moho depth to better fit the one of Brückl et al., which is apparently not the case. You should rather comment on that than on the accuracy of the 2 other models.

29) Discussion, l. 212-213: precise that Hrubcová et al. (2005) deals with the Bohemian massif.

30) Discussion, l. 215, 218: "latest at 300 km"? "anyways"? replace "one strong impedance" by "a strong impedance".

31) Discussion, l. 229-230: your GloPSI analysis fails to image the strongly dipping Moho resulting from the RF analysis at 400-550 km distance. You provide a number of possible explanations for that difference including the difficulty to image dipping boundaries with GloPSI or an anisotropic mid-lower crust. Why don't you firstly discuss the quality of the RF signals at these locations in Hetenyi et al. and also their migration model that you mention later in l. 253-254? As you are first author or co-author of the RF papers, you are the best expert to compare these results in more details.

32) Discussion, l. 255-256: comparison with the Western and Central Alps is useless as the geological context if different. You should erase the sentence "In accordance.. Alps" which does not provide any interesting information.

33) Discussion, l. 257: "a number of studies have proposed models of the deep structure beneath the Alps". You rather mean "beneath the Tauern window" or "beneath the high Eastern Alps east of 13°E" (because TRANSALP is in the E-Alps, and it can image the Moho).

34) Discussion, l. 262: correct "charachteristics".

35) Discussion, l. 263-265: do you really believe that the solution is in a better 3-D model from local earthquake tomography to improve the migration of RF, as suggested in your sentence "Obviously…across the plate boundary"? I don't. You cite the ANT study by Sadeghi-Bagherabadi et al. (2021) that uses data of the very dense Swath-D array. This paper shows a depth section along the EASI line where the Moho depth is computed from the Vs contours 4.1-4.3 km/s and compared to the RF Moho of Hetenyi et al. (2018). If these contours are a good proxy of the Moho, it is almost flat and continuous at 50 km depth in the Moho gap region where Hetenyi et al. propose 2 strongly dipping Moho surfaces. Although Sadeghi-Bagherabadi et al. has been published very recently, I would suggest that you mention this surprisingly simple result, in particular because it was computed using the densest 2-D array ever installed in the Alpine region. And because Swath-D exists, I don't think you can conclude that there is a need for increasing the station density in that region (last sentence).

36) Conclusion, l. 276: "..due to the southern dip of the European plate". Don't you rather mean "the southward dip of the European Moho"?

---

## Author Response (AR2)

*Dear Topical Editor,*
*We have implemented the changes suggested by you and the reviewer in our text.*
*We have listed here the comments and our replies, and we have prepared both a tracked-change and a clean version of the modified manuscript.*
* * *
Reviewer 1:

In particular, I suggest deleting from the main text most of the discussion on the comparison between GloPSI images obtained from data sets using 27 or 64 events and moving this discussion to supplementary materials. Also, I would suggest to better organize the discussion section by separating the discussion concerning the northern part of the profile and the one on the southern part. Currently, I find that there is too much back-and-forth between the two regions and between comparisons with previous studies.
*We have implemented the suggestions and corrections throughout the whole text.*

Hereafter I give more minor comments for each part of the manuscript.

Section 1: Introduction

L18: "plate Adria" -> "Adria plate"
*ok*

L19: "involved hundreds of kilometres, through […]" -> "involved hundreds of kilometres, of shortening, through […]"
*ok*

L27: It would have been interesting to mention / compare (in the discussion) the results obtained along the EASI profile with the ones along the TRNASALP transect to get insights on the lateral variations.
*The TRANSALP transect is located about 100km W of the EASI transect. While the former cuts across the western part of the Tauern window (TW), the latter passes E of the eastern border of the TW. From earlier but also a few recent studies (e.g. Rosenberg et al. 2017; Schmid et al. 2013 and references therein) the shallow crustal structure and tectonics of the western and eastern parts of the TW are known to be quite different, and the lateral variations and local details of the deep crustal structure beneath the TW region are emerging from the newer studies. Moreover, introducing the TRANSALP profile at this point would imply increasing the amount of reasoning and text in both the Introduction and Discussion {while you and the TE are encouraging to decrease the amount of text}, and there are several previous studies along the EASI line to compare our results with.*
*In consideration of all this we provide a comparative discussion of our results with the regional Moho topography but otherwise restrict the discussion to the immediate vicinity of the EASI transect.*

L41-44: This part (about the Moho depth estimates along the Cel09 and ALP75 CSS profiles) should be moved to the next paragraph.

*Moved*

L48: Why do you indicate "the needs to be interpolated" ? + Not clear what is the subject of "is" in "is interpretated with a Moho triple junction or a Moho gap".
*"that needs to be interpolated" has been deleted*

L49-52: "The later interpretation is strongly supported by Spada et al. […]": This sentence can be removed since your results do not support one interpretation or the other.
*ok*

L54: Add "phases" after "[…] with both Ps and Sp" + the word "scattered" does not seems appropriate since the conversions can simply be associated to various interfaces.
*ok*

L55-57: I would suggest removing "The interpretation of" since you are only summarizing the results of Hetenyi et al. (2018) and, later on, to remove "by different approaches" since this sentence is dealing with the results of the RF studies solely.
*Ok*

L63: Separate the 2 sentences + give more information on the results in terms of Moho depth (and lateral variations) based on ambient noise studies in the targeted area.
*Done according to TE's comment*

L67: Remove "new"
*ok*

L75-78: the sentence "In other implementations […]" is badly positioned (it should either be positioned before the previous sentence or in the next section) and, actually, I suggest removing it from the introduction
*Removed*

Section 2: Data and methods

L84-85: What is the influence of the "pure" P phases used? A supplementary figure showing the cross-section only with PKP and PKIKP phases would be useful to better see the impact of adding inline P-phases
*The impact of adding the P phases is hardly visible as we only found a few suitable events. For an array with a more favourable orientation with respect to illumination from events in the 30 to 90 deg distance range, it would be interesting indeed to show the image changing with adding illumination.*

L85-88: The sentence "We have used PKIKP […]" is just rephrasing the previous one and can be removed.
*ok*

L88-89: The discarded events are unclear. If you don't use events around 150°, you should

provide the exact distance range rejected. Otherwise, I don't understand the discarding of time windows in the sentence within the brackets.
*Modified according to TE's comment*

L90: The term "a high station coverage" is unclear here. Do you want to mention azimuthal coverage?
*The 64 events have been recorded at least by 80% of the stations, we clarified this in the text*

L98: No moveout correction is applied to account for the varying incident angles?
*Indeed, no move-out correction is needed. That is in fact the beauty of seismic interferometry, that from a range of illumination, through a process of constructive and destructive interference, the actual receiver-side reflections are isolated. We do not need any move-out correction since the correlation integral for GloPSI is well sampled near p=0, and it is near p=0 that most reflections exist (in a near horizontally stratified Earth).*
*Move-out correction is undesired since it adds reliance on an (always imperfect) model. If no illumination is used near p=0, no true zero-offset reflection response can be obtained. However, something that looks like a zero-offset reflection response can be obtained by applying a move-out correction (as is done in a few recent papers). Note that this processing has serious drawbacks since 1) the move-out correction adds another reliance on (an always imperfect) model and 2) P-S conversions are mapped to the move-out corrected gather. Numerical examples of obtaining a reflection response without any move-out correction needed can be found, e.g. in*
*http://homepage.tudelft.nl/t4n4v/4_Journals/Geophysics/geo_06c.pdf*

L103-108: The spectral balancing probably also aims at having an amplitude spectrum closer to the one of a dirac function to mimic an impulse response. I'm wondering if that can be added here.
*That is true. We have added this remark.*

L109-113: You can mention here the source-side reverberation acronym (SSR) used afterwards
*added*

L116: I don't understand the time range mentioned for the Hanning window (1 to 6 seconds) as the traces are much longer than 6s. Please clarify.
*This is applied for muting the delta pulse, as described in the text.*

L120-123: I would suggest giving a little bit more information on the process of multiple removing following Verschuur and Berkhout (1997) and the effect of this step. Based on the various figures (inc. the ones in the supplementary materials), the effect of this step is not evident and only seems to lower all the amplitudes in the signal (including reflectors within the crust).
*We agree that with the current visualisation the impact of the SRME is difficult to see (but for the lowering of the overall amplitude level which in the end is irrelevant). We had chosen to show the processing here without normalizing the maximum amplitude to 1 after each processing step (as, e.g. in Figure 2). This has a drawback that the removal of subtle*

*(multiple) reflections is not well seen. When flipping back and forth between normalized gathers it can be seen that the SRME attacks multiples especially in the northern part of the transect between 17 and 30 seconds.*
*We have added a line of explanation of the SRME implementation.*

Section 3: Results

L145-149: I suggest moving the small discussion of the effect of removing/adding clusters of events (Fig S8) to the supplementary text.
*done*

L155-156 + Fig. 3: It is weird to express standard deviations (std) in terms of % of the maximum amplitude in the window. This does not allow to clearly see the real std on the amplitude of the phases. Moreover, this relative scaling can be very dependent on spurious phases.
*This choice has been taken due to the comments of the reviewers in the first round of review. The STD as % of the maximum amplitude helps comparing the images in figure 3 and in figure S9. The STD is expressed as % of the maximum amplitudes in each panel.*

Fig 3: Question 1: Based on Fig. 3 it seems that the std is higher (including in the northern part) after the multiple suppression. Is it just an eye effect due to the change in the std scale of a real observation? If yes, why such step should increase the std (to my mind it should decreased it) ? Question 2: Why the std is smaller for times higher than 15s? Does it mean that the results are more stable in this time range (which I do not understand)?
*We changed the colorscale in order to show better that the std in the multiple suppression is lower than in the BAR and DPR images. For times after 15s there is little to no reflectivity therefore the amplitudes are very close to zero and the std is very low. We add this in the figure caption*

L159-167: I would suggest moving again this discussion between the GloPSI images with 27 and 64 phases to the supplementary materials.
*This is the description of Figure 4, and cannot be moved to the supplementary.*

L168-170: I am uncomfortable with the interpretation of the late spurious arrivals as P-S conversions. If you focus on the quasi-horizontal arrivals at depth higher than 60km, they correspond to phases with a lag time higher than 15-20s. If they are P->S conversions then they would occur within the mantle, which should be a less scattered medium than the crust. Alternatively, they would rather correspond to multiple crustal phases (PPS or PSS). Moreover, they are observed over a quite wide distance range, favoring horizontal structures. Why not simply considering that they can still represent SSR due to events present in both datasets (with 27 or 64 events)?
*Due to the different event distribution, the SSR should be different for the two images obtained by the pools of 27 and 64 events, therefore what we observe is RSR. It would be difficult to argue otherwise that increasing the amount of events does remove the SSR at the northern part of the profile, but not below the Alps.*
*We have added the alternative possibility of remnant multiples.*

L171: You should remove the last part of the sentence "[…] to focus our interpretation on the Moho topography in the northern part of the profile" since you actually discuss extensively later the southern part of the profile …
*done*

L180 and after: This section related to the possible interpretation of the scattered aspect of the image in the southern part would better fit into the discussion section. Indeed, you start here to discuss your results in comparison with other ones using different techniques (Brückl et al., 2007; Hetenyi et al., 2018) as you do in the next section.
*We deleted this part from the results section and added it with some modifications suggested by the Editor, at the end of the Discussion section*

Section 4: Discussion

L200-202: You are paraphrasing the last part of the previous section (see comment above).
*The last part of the previous section has been deleted*

L209-L213: The statement abouts the "accuracy" of past CSS studies (Brückl et al., 2007; Spada et al., 2013) should be reworded/strengthen. You consider your study as the reference, but you do not provide any estimate on uncertainties about your Moho depths either due to the data selection or (most importantly) to the velocity model (that can be different from the one used by these 2 former studies) used in your inversion. It would be better to use a term like "agreement" between your results and the previous ones.
*We modified according to comment #28 of the Topical Editor*

L221: I don't understand what you mean by "and the results of this latter"
*Hetenyi et al., 2018, modified*

L240: I would remove "[…] the reliably resolved" statement.
*removed*

L259: I don't understand why you say that the study from Yan and Mechie (1989) does not suffer from any method or data set limitations (compared to others) …
*Yan and Mechie (1989) report on the results obtained from the Alpine Longitudinal Profile ALP75. In a longitudinal refraction seismic profile where all shots yielded the necessary and expected seismic energy, the reversed imaged Moho sections are very reliably determined. Unfortunately, until today this remains the only such refraction seismic profile in the Eastern Alps. This is what we refer to. For more information on the 3D crustal modelling with (2D) CSS profile data see, f.e., Kissling, E., Ansorge, J. & Baumann, M. 1997. Methodological considerations of 3-D crustal structure modeling by 2-D seismic methods. In: Pfiffner , O. A., Lehner, P., Heitzmann, P., Mueller , S. & Steck, A. (eds) Deep Structure of the Swiss Alps. Birkhaeuser, 31–38.*

L262: characteristics
*modified*

L263: Why do you state that the Alpine orogen (in the targeted area) is characterized by "strongly dipping Moho interfaces"? (by the way I would remove the "s" from interfaces).
*this sentence has been removed following to the comments of the Topical Editor*

Section 5: Conclusion

L275-276: Please rephrase. The (potential) southern dip of the European plate is not the direct reason for delivering a clear image of the Moho with the GloPSI.
*Modified according to the Topical Editor's comment*

L277: I would replace "'inaccessible" by "unclear"
*done*

L278: Replace "this" by "the GloPSI"
*done*

Supplementary materials:

- Give more information on which base the 27 events (Fig S8) were selected
*done*
- Fig S2: There is a typo in the minimal distance
*done*
- Fig S7: I don't see the black dashed box mentioned in the legend
*figure modified*
- Table : Modify the figures referring to the 27 events (no more figure 2, Fig 4a and S8
*modified*
* * *
Topical Editor:

1) Abstract, l. 7: you write that your method "well images the topography of the Moho in regions where it shows a nearly planar behaviour… from the Bohemian massif to beneath the Northern Calcareous Alps". Is this really because the Moho topography is planar, or because it is reflective (corresponds to a strong velocity contrast)?
*modified*

2) Abstract, l. 9: what is a "typical" crust-mantle boundary?
*Modified by "evidence of the boundary"*

3) Abstract, l. 10: "absence of an Adriatic crust made of laterally continuous layers smoothly descending southwards". So, what is present? Is it a "structurally complex and faulted internal crustal structure" as suggested in the next sentence (but for the Alpine crust)?
*Yes, The two sentences are merged, so it is clearer.*

4) Abstract, l. 11: why do you conclude on a "structurally complex and faulted internal Alpine crustal structure". This contradicts the earlier sentence when you write that the Moho of the Northern Calcareous Alps is clear. They are not part of the Alps?
*You are right, we cancel the word "beneath" since beneath the Northern calcareous Alps the Moho is already not clear. The clear Moho is found in the Bohemian Massif to the Northern Calcareous Aps.*

5) Introduction, l. 16-18: useless details in sentence "After the closure of major and minor oceans, … continental parts of the much smaller plate Adria collided". Could be summarized to "After the closure of the Alpine Tethys, the European continental margin collided with the small Adria plate".
*We disagree with this and –as referenced– we follow the evolution concept by Handy et al. 2010 (and references therein). Between the European plate and the Adriatic plate, there was a complex series of small oceans and continental fragments, that in our view is the reason of such problematic deep structures in the Eastern Alps (east of 13E).*

6) Introduction: in fact, the previous comment is one example of an unnecessarily long and detailed sentence, and there are quite a few like that in the introduction. I am not sure that it is useful to cite all the experiments that have produced geophysical images in that region. You should focus the introduction on the key question that you address in that paper, which is the apparent Moho gap of Spada et al. (2013).
*Considering the conceptual relationship and differences in strength and limitations of the various seismic methods that were already applied to the region where we now apply the GloPSI method, we strongly believe that the information contained in the introduction are necessary. In the possible but not very likely event where the reader is fully aware of the details of all methods that were applied and all data sets that were obtained in the past about the lithosphere structure beneath the area traversed by the EASI transect, there would be no problem to even further reduce content and length of the introduction as we already did with the first revision. We prefer though to provide the background information and to give an overview on what has been done previously. Actually, the second part of this comment provides an example reason for this. The important question is not just about the presence or absence of the Moho gap in the area, the matter is why it is (and it has been in the previous studies) so hard to image the Moho in this section of the Alps, and why the several previous studies show disagreement.*
*As much as possible without deleting key information, we deleted some citations and some text and the remaining 2 pages of introduction seem appropriate in length.*

7) Introduction, l. 21-22: is there really "a general agreement that the European and the Adriatic Moho are offset across the plate boundary in the Alps"? Which publications state that?
*You are right, in particular due to the new publication by Sadeghi- Bagherabadi et al. (2021), we modified the sentence.*

8) Introduction, l. 23-44: I would suggest to avoid listing here all CSS experiments in the Alps and to keep only those related to the Eastern Alps.
*We deleted some citations and some text*

9) Introduction, l. 45-47: you write that most information about the Moho is derived from CSS experiments but you refer to publications such as Diehl et al. (2009) that only deals with earthquake sources. There is no contradiction, but this reference is inaccurate in this context.
*Deleted*

10) Introduction, l. 48: you should explain what the "Moho triple junction" of Brückl et al. is, because it is probably one of the questions that you want to address.
*Added*

11) Figure 1: you use similar thin plain lines (of different colors) to show very different features such as tectonic structures, the triple junction of Brückl et al. that refers to the Moho structure and to outline the area of Moho gap by Spada et al. This makes the figure confusing. I would suggest using different types of lines, following geological standards for the Alpine front for example and a filled polygon for the Moho gap area.
*done*

12) Introduction, l. 62-64: the sentence on recent ambient-noise tomography studies brings no useful information. I guess you mean that these ANT studies are more valuable for imaging velocity heterogeneities than imaging Moho depth variations. This is right, but it should be better explained. Moreover, some of the works you cite don't even reach Moho depth while others do and provide clues on the topography of velocity contours used as proxies for the Moho. This is worth mentioning.
*done*

13) Introduction, l. 69: rephrase unclear sentence ".. and stacking primarily global phases; waves that travel across the core…".
*done*

14) Introduction, l. 74: what do you mean by "considerably greater than zero"?
*Between 18 and 40 degrees. This has been added in the text.*

15) Introduction, l. 75: correct "Alpine reflectively". Do you mean reflectivity of structures of the Alpine crust?
*done*

16) Introduction, l. 75-76: sentence "In other…2019)" is out of context.
*cancelled*

17) Section 2.1, l. 88-89: did you discard entire event recordings or did you only discard time windows with multiple phases? Please rephrase.
*cancelled*

18) Section 2.2, l. 97: please rephrase "selecting minus the causal result and muting the delta pulse".
*We extended this part with easier-to-understand wording*

19) Section 2.2, l. 106: I guess "rupture effects" means "earthquake source effects".
*done*

20) Section 2.2, l. 112: by "reflectivity from the lithosphere at the source", you probably mean "spurious signals from the lithospheric structure at the source side".
*ok*

21) Section 2.2, l. 114: step without "s"
*ok*

22) Results, l. 161: replace "especially receiver-side reflectivity is shown on these images" by "these images mostly show receiver-side reflectivity"
*done*

23) Results, l. 171: you write that you decide "to focus (your) interpretation on the Moho topography in the northern part of the profile". This is surprising at this step of the paper because the most interesting objective is the "Moho gap" in the southern end.
Do you mean that you quickly give up on bringing in new constraints on the most interesting southern part, and that you will not discuss this part further?
*Not at all, as the reviewer says: "L171: You should remove the last part of the sentence "[…] to focus our interpretation on the Moho topography in the northern part of the profile" since you actually discuss extensively later the southern part of the profile". We do discuss the reasons why the Moho signal is not clear in our image in the Discussions*

24) Results, l. 182: by "suggest the signals representing at least in parts internal crustal structure", do you mean that the amplitude difference between signals at crustal depth in the northern and southern parts suggests that at least part of the signals in the south side can be attributed to actual crustal structure?
*yes*

25) Results, l. 182-185: The sentence "Unfortunately, the 3D crustal structure of the Eastern Alps below 15 km depth is still poorly known … with reference to the tectonic style and geologic evolution of the orogeny (e.g. Willingshofer et al., 2013; Rosenberg and Kissling, 2013, and references therein)" is too long and unclear, and it is partly wrong. I would consider that the crustal structure of the Eastern Alps, with TRANSALP and EASI, has been studied by as many tomography studies as the Western Alps with the CIFALPS profiles and ECORS-CROP. The crustal structure of the Central Alps is more poorly know since it has only been studied by the NFP-20 deepseismic sounding profiles, and no dense passive seismic experiment. The reference that you give (Kissling et al., 2006) presents a synthesis of what was known at the time of writing, that is before a number of recent experiments in the Western and Eastern Alps, including EASI. You should update your reference list. I don't know Behm et al. (2006) which is an unpublished

PhD thesis whose citation is useless. You also cite Lu et al. (2020) that covers the entire Alps, and not only the Western and Central Alps, and provides the Vs structure at depth >15 km in contradiction with your sentence. Qorbani et al. (2020) does cover only the Eastern Alps to ~40 km depth, also in contradiction with your sentence. Molinari et al. (2020) and Sadeghi-Bagherabadi et al. (2021) also focus on the crustal structure of the Eastern Alps. That's a lot of publications on the crustal structure of the E-Alps in the end! The problem of the lack of clear images of the structure of the lower crust and Moho beneath the Tauern window is obviously not due to the lack of data. I don't understand what you mean by "and with reference to the tectonic style and geologic evolution of the orogeny". Please clarify.

*We delete these sentences from the Results section (as suggested by the reviewer) and acknowledge the newer data and studies at the end of the Discussion section.*

26) Results, l. 185-186: In the next sentence, you write that you expect a complex crustal structure and you cite a review paper (Handy et al., 2015) that deals with palinspatic reconstructions and slab geometry. Again, a tomography paper that shows that imaging the lower crust is particularly difficult beneath the Tauern window, like Hetenyi et al. (2018) is more adequate. You should maybe erase these 2 sentences and leave only the one of l. 187-189, which is much more correct and accurate.

*As stated before these sentences are deleted*

27) Discussion, l. 207-210: When you write "the strength of (your) new results lies in the continuous assessment of the lateral variation of the Moho interface… in the northern part of the profile", you seem to forget the RF results of Hetenyi et al. (2018) who were the first to provide a continuous image of the depth variations of the Moho beneath the same profile. This is surprising as the first author of the present paper is a co-author of Hetenyi et al. (2018). You should start the discussion by comparing with their results. This sentence is also contradictory with the one of l. 224 "we conclude the Moho is well imaged univocally by all methods in this northernmost section". If all methods work well in that part of the profile, imaging the same Moho as others cannot be the strength of your new results.

*We deleted that sentence*

28) Discussion, l. 210: You cannot tell that the Moho model of Spada et al. is more accurate than the one by Brückl et al. only because the first one better fits your Moho depth estimate. The three Moho depth models depend on the velocity models used to convert time to depth. You use the Vp model by Brückl et al. shown in Fig. S10. I would therefore expect your Moho depth to better fit the one of Brückl et al., which is apparently not the case. You should rather comment on that than on the accuracy of the 2 other models.

*We modified according to this suggestion*

29) Discussion, l. 212-213: precise that Hrubcová et al. (2005) deals with the Bohemian massif.

*Now it's explicit*

30) Discussion, l. 215, 218: "latest at 300 km"? "anyways"? replace "one strong

impedance" by "a strong impedance".
*done*

31) Discussion, l. 229-230: your GloPSI analysis fails to image the strongly dipping Moho resulting from the RF analysis at 400-550 km distance. You provide a number of possible explanations for that difference including the difficulty to image dipping boundaries with GloPSI or an anisotropic mid-lower crust. Why don't you firstly discuss the quality of the RF signals at these locations in Hetenyi et al. and also their migration model that you mention later in l. 253-254? As you are first author or co-author of the RF papers, you are the best expert to compare these results in more details.
*Few comments have been added*

32) Discussion, l. 255-256: comparison with the Western and Central Alps is useless as the geological context if different. You should erase the sentence "In accordance.. Alps" which does not provide any interesting information.
*Erased*

33) Discussion, l. 257: "a number of studies have proposed models of the deep structure beneath the Alps". You rather mean "beneath the Tauern window" or "beneath the high Eastern Alps east of 13°E" (because TRANSALP is in the E-Alps, and it can image the Moho).
*modified*

34) Discussion, l. 262: correct "charachteristics".
*modified*

35) Discussion, l. 263-265: do you really believe that the solution is in a better 3-D model from local earthquake tomography to improve the migration of RF, as suggested in your sentence "Obviously…across the plate boundary"? I don't. You cite the ANT study by Sadeghi-Bagherabadi et al. (2021) that uses data of the very dense Swath-D array. This paper shows a depth section along the EASI line where the Moho depth is computed from the Vs contours 4.1-4.3 km/s and compared to the RF Moho of Hetenyi et al. (2018). If these contours are a good proxy of the Moho, it is almost flat and continuous at 50 km depth in the Moho gap region where Hetenyi et al. propose 2 strongly dipping Moho surfaces. Although Sadeghi-Bagherabadi et al. has been published very recently, I would suggest that you mention this surprisingly simple result, in particular because it was computed using the densest 2-D array ever installed in the Alpine region. And because Swath-D exists, I don't think you can conclude that there is a need for increasing the station density in that region (last sentence).
*The last sentences of the "Discussion" have been modified.*

36) Conclusion, l. 276: "..due to the southern dip of the European plate". Don't you rather mean "the southward dip of the European Moho"?
*modified*